# Single-cell RNA sequencing reveals the mesangial identity and species diversity of glomerular cell transcriptomes

Bing He [1,11], Ping Chen [1,11], Sonia Zambrano [1,2], Dina Dabaghie[1,2], Yizhou Hu [3], Katja Möller-Hackbarth[1], David Unnersjö-Jess [4,5], Gül Gizem Korkut[1], Emmanuelle Charrin[1,2], Marie Jeansson [1], Maria Bintanel-Morcillo[1,2], Anna Witasp[6], Lars Wennberg[7], Annika Wernerson[6], Bernhard Schermer[4,5], Thomas Benzing [4,5], Patrik Ernfors [3], Christer Betsholtz [1,8], Mark Lal [9], Rickard Sandberg [10] & Jaakko Patrakka [1,2✉]

Molecular characterization of the individual cell types in human kidney as well as model organisms are critical in defining organ function and understanding translational aspects of biomedical research. Previous studies have uncovered gene expression profiles of several kidney glomerular cell types, however, important cells, including mesangial (MCs) and glomerular parietal epithelial cells (PECs), are missing or incompletely described, and a systematic comparison between mouse and human kidney is lacking. To this end, we use Smart-seq2 to profile 4332 individual glomerulus-associated cells isolated from human living donor renal biopsies and mouse kidney. The analysis reveals genetic programs for all four glomerular cell types (podocytes, glomerular endothelial cells, MCs and PECs) as well as rare glomerulus-associated macula densa cells. Importantly, we detect heterogeneity in glomerulus-associated *Pdgfrb*-expressing cells, including bona fide intraglomerular MCs with the functionally active phagocytic molecular machinery, as well as a unique mural cell type located in the central stalk region of the glomerulus tuft. Furthermore, we observe remarkable species differences in the individual gene expression profiles of defined glomerular cell types that highlight translational challenges in the field and provide a guide to design translational studies.

[1] Karolinska Institute/AstraZeneca Integrated Cardio Metabolic Center (ICMC), Huddinge, Sweden. [2] Division of Pathology, Department of Laboratory Medicine, Karolinska University Hospital Huddinge, Huddinge, Sweden. [3] Division of Neuroscience, Department of Medical Biochemistry and Biophysics (MBB), Karolinska Institute, Stockholm, Sweden. [4] Department II of Internal Medicine and Center for Molecular Medicine Cologne (CMMC), University of Cologne, Cologne, Germany. [5] Cologne Excellence Cluster on Cellular Stress Responses in Aging-Associated Diseases (CECAD), University of Cologne, Cologne, Germany. [6] Division of Renal Medicine, Department of Clinical Sciences, Intervention and Technology, Karolinska Institute, Stockholm, Sweden. [7] Division of Transplantation, Department of Clinical Sciences, Intervention and Technology, Karolinska Institute, Stockholm, Sweden. [8] Department of Immunology, Genetics and Pathology, Uppsala University, Uppsala, Sweden. [9] Bioscience, Cardiovascular, Renal and Metabolism, R&D Biopharmaceuticals, AstraZeneca Gothenburg, Sweden. [10] Department of Cell and Molecular Biology, Karolinska Institute, Stockholm, Sweden. [11] These authors contributed equally: Bing He, Ping Chen. ✉email: Jaakko.patrakka@ki.se

The glomerular capillary tuft responsible for the kidney ultrafiltration is composed of three cell types: podocytes, glomerular endothelial cells (GECs) and mesangial cells (MCs). Podocytes are highly specialized, terminally differentiated epithelial cells[1], whereas MCs are considered to be glomerular pericytes[2]. Glomerular parietal epithelial cells (PECs) lining Bowman's capsule of the glomerulus are an integral part of the filter unit and speculated to act as stem cells for both podocytes and proximal tubular cells (PTCs)[3]. Together, these glomerular cells form a micro-organ that is functionally and structurally unique in the body, and therefore expected to have a highly specialized molecular composition. However, so far only the molecular identity of the podocyte has been extensively characterized, which has been accomplished by using cell-specific reporter mouse lines. The studies on mouse GECs and MCs have been more limited, mostly due to lack of cell-type-specific genes enabling the generation of reporter lines, whereas the molecular architecture of PECs is almost completely unknown.

Recently, the introduction of single-cell RNA sequencing (scRNA-seq) technologies has launched a new era in studing the molecular profiling of complex and heterogeneous tissues. In the adult kidney, published scRNA-seq studies have mostly been performed using Drop-seq/10xGenomics-based platforms[4–8], enabling the capture of a great number of cells at a relatively limited sensitivity in gene detection compared with, for example, the Smart-seq2 platform[9]. In a recent scRNA-seq study on isolated mouse glomeruli, transcriptome profiles of podocytes, GECs and cells that were annotated as MCs were described[5] and molecular markers for specific cell types, as well as previously undetected heterogeneity in GECs were also presented. A validation of the anatomical position of the cells described as MCs and evidence that these cells represent the unique cell type intrinsic to the glomerular tuft was however missing. Another scRNA-seq study on human whole kidneys reported podocyte transcriptomes but lacked properly annotated MCs and GECs[6].

Glomerular disease processes are the major cause of end-stage renal disease[10]. Glomeruli can be damaged by systemic diseases, such as in diabetic nephropathy (DN), or by a primary glomerular insult, such as in IgA nephropathy. In most human glomerulopathies, MCs are activated and proliferate, which correlates with the loss of kidney function[11,12]. Similarly, podocyte damage and loss play a key role in disease pathogenesis[13,14]. Because little is known about the molecular mechanisms behind these events in humans, the mouse is used widely to model human glomerulopathies. However, translational studies have been challenging. For instance, different mouse models of DN recapitulate poorly the molecular mechanisms and histological features detected in human diabetic glomeruli[15]. Similarly, injection of serum from patients with membranous nephropathy (MN) containing anti-phospholipase phospholipase A2 receptor (PLA2R1) auto-antibodies targeting podocyte foot process to mice has failed to recapitulate the main features of this human disease[16]. On the other hand, as gene manipulation is relatively easy in mouse, it will likely remain as the most important animal to model human glomerular disease processes. Thus, a bridge connecting the gap between two species is required for translational studies.

To gain insights into the molecular signature of the glomerular cells and translational aspects of target validation, we performed scRNA-seq on human and mouse glomeruli using the Smart-seq2 platform. We identified all glomerular cell types including PECs and importantly, defined the bona fide transcriptional signature of glomerular tuft-intrinsic MCs. We further identified a unique mural cell type located in the stalk/root of the glomerulus tuft. These cells showed a molecular signature distinct from the glomerular tuft-intrinsic MCs, suggesting a specialized function. Comparative analyses between mouse and human glomerular cells revealed conserved expression of the vast majority of cell-specific genes but also identified a number of genes uniquely expressed in the human vs mouse, including for instance PLA2R1, a human podocyte-specific expressed gene that is absent in the mouse. Our results define the precise cellular identity of MCs and unravel unexpected species diversity among glomerular transcriptional signatures, insights essential for translational studies.

## Results

**Glomerular cell clustering and annotation**. To investigate cell types and expression patterns in mouse and human kidney glomeruli, we dissociated the isolated glomerular tissues followed by FACS single-cell sorting, and then subjected them to Smart-seq2, a scRNA-seq method that is more sensitive in gene detection compared to droplet-based methods[9] (Fig. 1a). Unsupervised clustering identified 16 and 9 clusters of cells in mouse and human samples, respectively (Fig. 1b and Supplementary Fig. 1a, b). Based on known cell-type markers (Fig. 1c and Supplementary Fig. 1c–f), these cell clusters were assigned to distinct cell types, including three principal glomerular cell types: *Nphs1*+/*NPHS1*+ podocytes, *Pecam1*+/*PECAM1*+ ECs, *Pdgfrb*+/*PDGFRB*+ cells (designated here as mesangial-like cells, MLCs), and glomerulus-associated *Ptprc*+/*PTPRC*+ immune cells and tubular cells. In addition, we identified a small population of glomerular PECs expressing *Cldn1* and *Pdgfrb* (Fig. 1c), described later in detail. The majority of human *NPHS1*+ podocytes expressed *PDGFRB*, but only a few of mouse podocytes expressed *Pdgfrb*. The captured mouse and human immune cell populations were further in silico confirmed by mapping these single cells to the published human kidney-associated immune cell atlas[17] (Supplementary Fig. 2). We observed two distinct subclusters of mouse *Pecam1*+ ECs, whereas human *PECAM1*+ ECs were homogeneous (Fig. 1b). Further analysis demonstrated the two subsets in mouse represented *Ehd3*- and *Kdr*-expressing GECs and *Fbln5*- and *Glul*-expressing extra-glomerular ECs (Supplementary Fig. 3 and Supplementary Data 1–3), suggesting the inclusion of afferent/efferent arterioles during mouse glomerulus isolation. The numbers of genes detected per cell and per cell type are summarized in Supplementary Fig. 4.

Our annotated cell types could also be independently defined by active transcription factor (TF) regulatory modules on the basis of scRNA-seq data (Supplementary Fig. 5), especially for the three principal glomerular cell types (Fig. 1d). Several cell-type-specific TF modules were active in both mouse and human, such as *Erg*, *Gata2*, *Gata5*, *Sox18*, *Rarg*, *Pbx1* and *Bcl6b* for GECs, *Wt1* for podocytes and *Gata3* for MLCs. Interestingly, *NR1H3* and *Nr2f2* identified in human and mouse MLCs, respectively (Fig. 1d), have been shown to be important for the regulation of renin expression[18,19]. These two TFs may be linked to the transcriptional signature of renin lineage cells. On the other hand, active TF modules were not able to clearly distinguish subpopulations within cell types, such as distinct distal convoluted tubular (DCT) cell and immune cell subpopulations in mouse (Supplementary Fig. 5). A searchable database of our Smart-seq2 single-cell gene expression data is available here: https://patrakkalab.se/kidney.

**The molecular signature of MLCs**. As *Pdgfrb* is thought to be expressed in all mural cells including vascular smooth muscle cells (vSMCs), pericytes and even fibroblasts[20], we speculated that like the case of GECs, the MLCs may contain other cell types in addition to intraglomerular MCs. To address this question, more MLCs were obtained by sorting glomerular cells from six additional mice through the exclusion of CD45+ and CD31+ cells. All

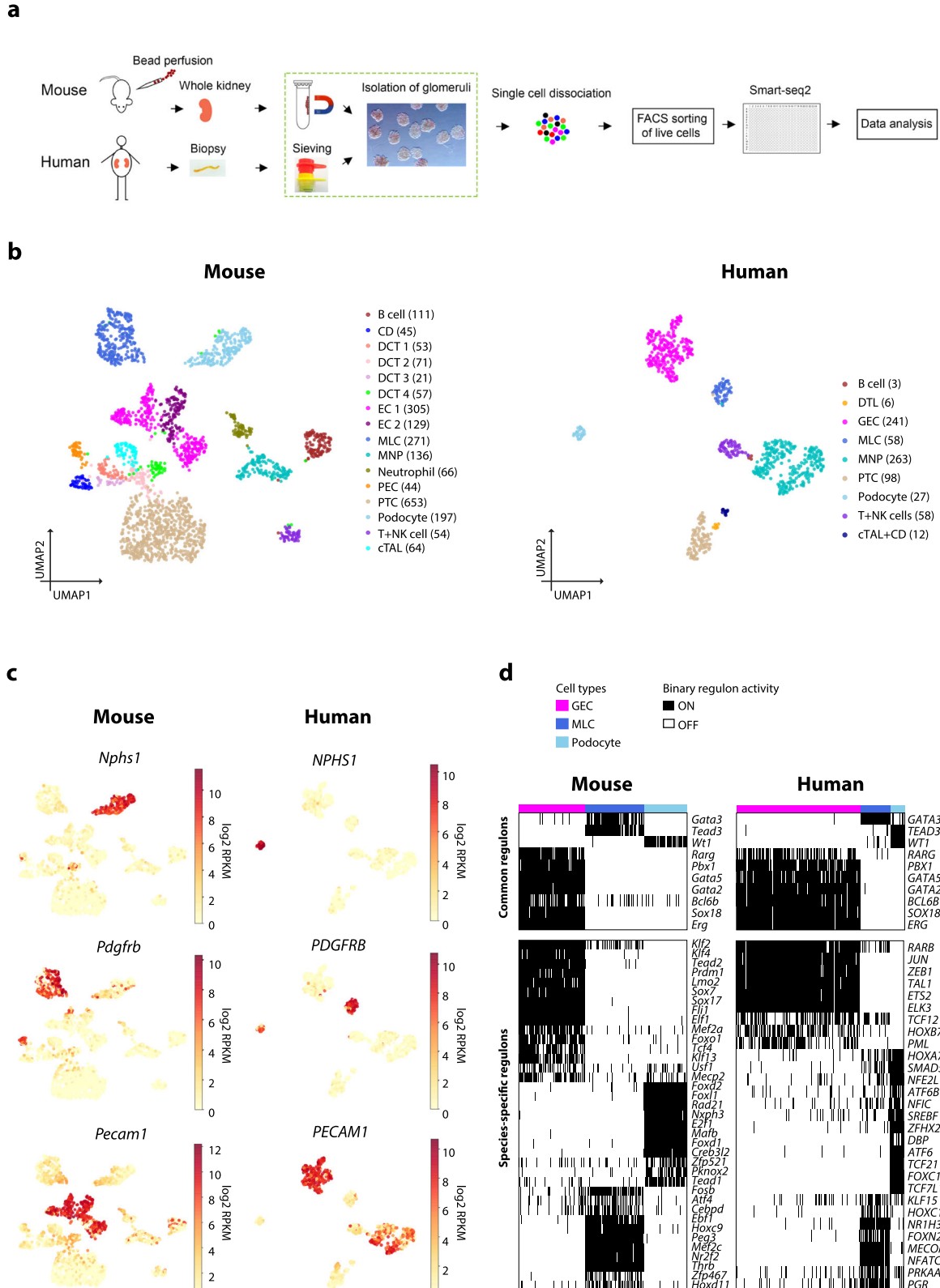

mouse *Pdgfrb⁺ Cldn1⁻* cells by this biased sorting (Supplementary Fig. 6a–c) and by unbiased sorting (Fig. 1b) were pooled and re-analysed as MLCs (*n* = 339, Fig. 2a and Supplementary 7a). Subclustering analysis revealed four distinct subpopulations within MLCs (Fig. 2a), demonstrating differential expression of classical vSMC, pericyte and fibroblast marker genes (Fig. 2b and

Supplementary Data 4). MLC-C2 and MLC-C3 showed high expression of vSMC markers, such as *Cnn1* and *Acta2*, and MLC-C3 was also strongly expressing *Ren1* (Supplementary Data 4 and 5). Thus, MLC-C2 and MLC-C3 represented a mixture of vSMCs and renin-producing cells, originating most likely from glomerulus-associated afferent/efferent arterioles.

**Fig. 1 scRNA-seq analysis of isolated mouse and human glomerulus-associated cells. a** Summarized study design and experimental procedure. **b** Projection of mouse and human single cells onto 2-D UMAP space coloured by assigned cell types. Cell numbers in each population are shown in parenthesis. **c** The expression of known cell type marker genes in mouse and human principal glomerular cells. The markers include *Nphs1/NPHS1* for podocytes, *Pdgfrb/PDGFRB* for mesangial-like cells (MLCs) and *Pecam1/PECAM1* for ECs. The colour intensity of dots represents the expression level. The colour scale is defined by log$_2$(mean RPKM). **d** Mouse and human podocyte, glomerular EC, and MLC enriched active regulons constructed by SCENIC. Regulons active in >50% of cells in one cell type but not in other two cell types were considered as cell-type enriched regulons. Binary activity scores are shown in the heatmap with black indicating 'ON/active' and white indicating 'OFF/inactive'. CD collecting duct, DCT distal convoluted tubule, PTC proximal tubule cell, EC endothelial cell, MLC mesangial-like cell, MNP mononuclear phagocyte, PEC glomerular parietal epithelial cell, cTAL cortical thick ascending limb; colour dots indicate diverse cell types.

Interestingly, the fibroblast marker gene *Pdgfra* was exclusively expressed by MLC-C4 (Fig. 2b and Supplementary Data 4). In addition, *Gata3* was expressed highly by MLC-C2, MLC-C3 and MLC-C4, but was absent in MLC-C1 (Fig. 2b). To define the identity of MLC-C4 cells, we analysed the expression of key marker genes at the protein level using a reporter mouse line and immunostaining assays. Intraglomerular desmin-positive MCs were labelled by the *Pdgfra*-H2BGFP[21], whereas extraglomerular vSMCs were not (Fig. 2c). This was further confirmed in human kidney in which PDGFRA was detected only in intraglomerular MCs and calponin-1, encoded by *CNN1*, solely in vSMCs outside the glomerulus (Supplementary Fig. 7b). Thus, MLC-C4 cells represented bona fide intraglomerular MCs. Of note, in human almost all *PDGFRB*$^+$ cells exhibited a similar profile as mouse MLC-C4 (Supplementary Fig. 7c, d), indicating that they represented intraglomerular MCs.

To reveal the identity of MLC-C1, triple labelling staining for key differentially expressed genes was performed in mouse kidney. We detected a PDGFRB$^+$ cell group located in the stalk/root of the glomerular tuft that was positive for aSMA but negative for calponin-1 (Fig. 2d). Similarly, in line with our scRNA-seq data, this aSMA/PDGFRB-positive cell population was negative for GATA3 (Fig. 2e). Thus, MLC-C1 cells represented a distinct glomerulus-associated mural cell population that was located in the stalk/root region of the glomerular tuft, that corresponds probably to previously vaguely described Lacis or extraglomerular mesangial cells (EMCs)[22,23].

Importantly, all four MLC subpopulations exhibited distinct transcriptome profiles (Fig. 2f and Supplementary Data 4). To define in more detail the putative EMC population, we compared the molecular signatures between EMCs and MCs (Supplementary Fig. 8 and Supplementary Data 6). Using a fold change of 3.0 (median log$_2$-transformed RPKM, adjusted $P < 0.05$) as a cut-off, 69 genes were significantly upregulated in EMCs and 20 genes in MCs (Supplementary Data 6 and Supplementary Fig. 8). We noticed that many vSMC-associated genes, such as *Tagln*, *Myh11*, *Acta2*, *Notch3* and *Ednra*, were among in EMC-upregulated genes. They also included *Fxyd1* that has been reported as a marker of EMCs[23], supporting the annotation of MLC-C1 as EMCs. In addition, several genes (*Exph3*, *Sod3*, *Cygb* and *Olfr78*) showed specific upregulation in EMCs compared to all other MLC clusters (Fig. 2f and Supplementary Data 4). Thus, MLC-C1 (EMCs) has a very distinct molecular profile from MCs, although they both have been classified as MCs residing in two contiguous compartments.

Next, to better understand the biology of MLC subpopulations, we in-silico compared the transcriptomes of these cells to well-defined pericytes, vSMCs, fibroblasts and other control cell types using previously published scRNA-seq data from mouse lung[24]. All MLC clusters showed similarities to vSMCs according to estimated probabilities (Fig. 2g, h), but MLC-C2 and MLC-C3 reached highest similarities to vSMCs. Interestingly, human and mouse MCs showed higher similarities to pericytes and fibroblasts than other MLC subpopulations (Fig. 2g, h). The cell

type similarity of MLC-C1 to the reference cell types appeared to be between MCs and vSMCs, implying a transitional status. Probabilistic similarities did not occur by chance as randomized control cells did not show a similarity towards any cell type (Supplementary Fig. 9), suggesting that MCs possess a complex phenotype with features of pericytes, vSMCs and fibroblasts.

To validate identified characterization of the mesangial identity and MLC heterogeneity, we generated an independent scRNA-seq data by specifically sorting glomerular EGFP-labelled PDGFRB$^+$ cells from a *Pdgfrb*-EGFP reporter mouse line. This data replicated the presence of the distinct four subsets of *Pdgfrb*$^+$ MLCs (Supplementary Fig. 10). Interestingly, these MCs captured from two mouse lines (C57BL/6J and PDGFRB$^+$) and/or from the human shared active TF regulatory modules in lung pericytes, vSMCs and fibroblasts (Fig. 2I). For example, *Ets1* and *Atf3* regulons were specifically active in pericytes and vSMCs, respectively, whereas *Fos* regulon was active in both vSMCs and fibroblasts. This implies that the complex phenotype of MCs may be co-regulated by cell type determinants in vSMCs, pericytes and fibroblasts.

To test whether the same heterogeneity of glomerulus-associated mural cells was present in other reported single-cell studies, we re-analysed the previously published scRNA-seq data that described mouse MCs[5]. Our analysis showed that this group of cells had clearly four different subpopulations: (1) cells showing strong podocyte signature, representing probably contaminated cells; (2) renin-expressing cells, representing cells of the juxtaglomerular apparatus (JGA); (3) cells showing high expression of vSMCs markers, probably representing vSMCs from afferent/efferent arterioles; (4) *Pdgfrb*$^+$ cells showing weaker vSMCs-like signature that may represent true MCs (Supplementary Fig. 11). Taken together, it seems that we captured transcriptional profiles of genuine MCs.

**Identification of phagocytic activity in MCs.** To get more insights into the molecular machinery of MCs, we performed pathway analysis of genes enriched in MCs (vs other MLCs). Interestingly, this unravelled several pathways involved in phagocytosis (Supplementary Data 7). This was of special interest as MCs have been speculated to be active in phagocytosis[25], but the data are controversial. Moreover, we built up a MC-centric ligand–receptor interaction atlas between three principal glomerular cell types (Fig. 3a, b and Supplementary Fig. 12a, b), and observed a crosstalk pathway, in which LDL receptor-related protein 1 (LRP1) on MCs potentially interacted with F8 and APP from GECs. LRP1 is a multifunctional scavenger involved in phagocytosis[26] and its expression in mouse and human MCs is high, but notably low in vSMCs and renin cells (Supplementary Data 5). LRP1 expression in mouse MCs was validated by immunostaining (Supplementary Fig. 13a).

To functionally validate the presence of phagocytic activity in MCs, we performed two in vitro and one in vivo assay. In mouse, we isolated fresh glomerular cells from the *Pdgfrb*-EGFP mice and

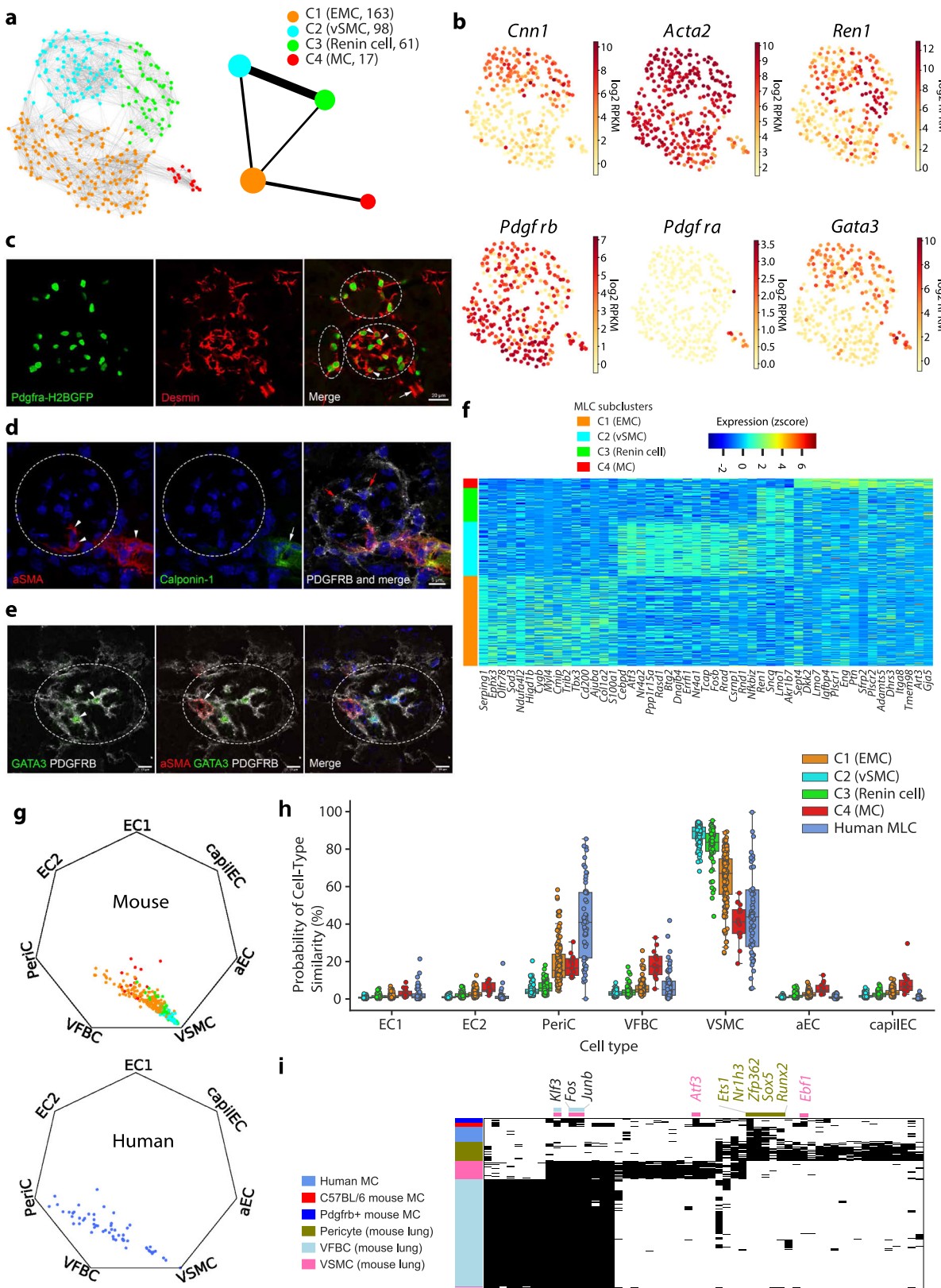

FACS sorted EGFP⁺ cells for the analysis. The cells were incubated with latex beads conjugated with a pH-sensitive fluorescence and the results were analysed by FACS. In the assay, the baseline signal of bead⁺ cells at 4 °C was very low as only one out of 212 cells (0.5%, $n = 3$) was positive (a flow cytometry scatter plot for one out of three assays shown in Fig. 3c,

summarized data in Fig. 3d). EGFP⁺ glomerular cells demonstrated a robust activation of phagocytosis at 37 °C as 63.7% of cells (1508/2368, $n = 3$) showed positive signal (Fig. 3c, d). Uptake of beads by mouse glomerular EGFP⁺ cells was further visualized by a confocal microscopy analysis, which showed co-localization of fluorescent beads with EGFP⁺ glomerulus-

**Fig. 2 The heterogeneity of mouse *Pdgfrb*-expressing MLCs. a** Trajectory of all mouse *Pdgfrb*-expressing MLCs by UMAP (left) and partition-based graph abstraction (right). MLCs were classified into four subclusters (MLC-C1–C4). Batch effect correction was performed as the cells were generated from two independent experimental batches using unbiased and CD45⁻ CD31⁻ sorting. In the PAGA graph, each node represents one MLC subcluster and the edge weights quantify the neighbourhood relation. The node size indicates the number of cells in each subcluster. EMC extraglomerular cell, vSMC vascular smooth muscle cell, MC mesangial cell. **b** The expression (log$_2$-transformed RPKM) of classical marker genes for vSMCs (*Cnn1*, *Acta2*), renin cells (*Ren1*), pericytes (*Pdgfrb*), fibroblasts (*Pdgfra*) and *Gata3* visualized in UMAP. The colour scale is defined by log$_2$(mean RPKM). **c** The mesangial localization of PDGFRA in the mouse glomerulus. The staining for desmin (red) localizes to PDGFRA-positive cells (green, arrowheads) in glomeruli in the *Pdgfra*-H2BGFP reporter mouse line. Desmin-positive vSMCs (arrow) outside glomeruli show no GFP signal for PDGFRA. White discontinuous circles indicate glomeruli. Scale bar: 20 µm. **d** Triple labelling for PDGFRB, Calponin-1 and aSMA in the mouse kidney. Calponin-1 (green) is detected only in afferent/efferent arteriolar SMCs (white arrow), whereas aSMA (red) is detected in both vSMCs and PDGFRB (white)-positive cells located in the central stalk of the glomerulus (white arrowheads). PDGFRB is also positive in intraglomerular MCs (red arrows). Scale bar: 5 µm. **e** Triple labelling for PDGFRB, aSMA and GATA3 in the mouse kidney. Nuclear location of GATA3 (green) is detected in PDGFRB (white)-positive intraglomerular MCs (arrowheads), but not in double aSMA (red) and PDGFRB (white)-positive cells (arrows) located in the central stalk of the glomerulus. Scale bar: 10 µm. **f** Heatmap showing the top 15 genes significantly upregulated in each MLC subcluster. The genes were selected based on the magnitude of expression range among four MLC subclusters. A full list of differentially expressed genes is presented in Supplementary Data 4. **g** Radar visualization of the probabilistic similarity of mouse and human MLCs in relation to defined pericytes, vSMCs, fibroblasts and various EC subpopulations from mouse lung. **h** The boxplot showing the probabilistic similarity of MLCs (*y*-axis) to reference cell types (*x*-axis) binned by MLC subclusters. The estimation was done on $n = 339$ mouse and $n = 58$ human MLCs over $n = 1209$ reference cells from mouse lung. The box plot illustrates the first quartile, median and the third quartile with whiskers of maximum 1.5 IQR (the interquartile range). **i** The activity of mouse lung pericyte, vFBC and vSMC active regulons ($n = 57$) in mouse and human glomerular MCs. Binary activity scores are shown in the heatmap with black indicating 'ON/active' and white indicating 'OFF/inactive'. Regulators active in MCs from two mouse strains and/or in human MCs are highlighted with colour matching to mouse lung cell types.

associated and individual EGFP⁺ cells (Fig. 3e). The assay was validated by analysing bead⁺ leucocytes isolated from wild-type (wt) mouse blood that showed the induction of phagocytosis at 37 °C (Supplementary Fig. 13b).

To see whether this occurred also in human MCs, we isolated glomeruli from human kidneys and performed in vitro phagocytosis assay in PDGFRB⁺ cells. When primary human glomerular cells staining positive for PDGFRB were incubated with conjugated latex beads, a total of 21.7% (108/498, 17.3–30.1%, $n = 3$) of cells showed intracellular beads (Fig. 3f). This result suggests that also human MCs can perform phagocytosis in cell culture (similarly to mouse MCs)

Finally, to analyse whether particle-uptake by MCs could occur in vivo, we injected mice with FITC-labelled bovine serum albumin (BSA) and followed the accumulation of fluorescent signal in kidney glomeruli after the injection using the super-resolution STED microscopy[27]. One hour after the injection, BSA was detected in the glomerular basement membrane (GBM) indicating that it passed freely through GEC layer and reached glomerular matrix (Fig. 3g). More interestingly, FITC-labelled BSA was accumulating intracellularly in PDGFRB⁺ MCs, suggesting an active uptake of BSA in vivo. Quantitatively, a total of 271 out of 389 analysed PDGFRB⁺ cells (70.0%, 67.1–82.8%, $n = 3$) showed intracellular FITC, whereas no signal was detected in uninjected animals. This in vivo data, together with our in vitro data and the identified molecular signature of MCs, strongly argues for the presence of significant phagocytic activity in MCs.

**Species diversity of principal glomerular cell transcriptomes**. We performed cross-species comparisons on the transcriptomes of the three principal glomerular cell types (podocytes, MCs and GECs), respectively (see Supplementary Fig. 14 for the overall approach). For each cell type, genes conservatively expressed at high level and genes specifically expressed in one species were highlighted with colour in a scatter plot; conserved cell-type-specific gene expression was visualized in dot plots; species-specific gene expression was illustrated in heatmaps (Figs. 4–6). To validate candidates of interest, we performed in silico validations using multiple data cohorts and experimental validations using RT-PCR, immunostaining/Western blot.

**Mesangial cells**. The definition of MCs allowed us to make a comparison of homogeneous MC transcriptomes between mouse and human. We observed that 243 genes were highly expressed in MCs from both species (Fig. 4a and Supplementary Data 8). Among the conserved genes, only four genes (*Gata3*, *Pdgfra*, *Itga8* and *Sept4*) appeared to be specific to MCs across all annotated cell types (Fig. 4b). Clearly, expression of *Pdgfra* was the most specific across glomerular and tubular cells, supporting it as a good MC marker. A total of 75 MC-expressed genes showed significant species-specific expression patterns between mouse and human (Fig. 4a, c). Due to unavailability of either published bulk or scRNA-seq data from both species for this pure cell population, we chose a mouse-specific gene *Des* commonly used as a MC marker and a human-specific gene *COL6A1*, encoding a poorly characterized glomerular extracellular matrix member for experimental validation. Immunohistochemically, desmin was clearly expressed in mouse MCs but completely absent in the human glomerulus (Fig. 4d). Similarly, in line with our scRNA-seq data, human COL6A1 was robustly expressed in human MCs, whereas mouse COL6A1 was almost completely absent in PDGFRB-marked MCs in double immunofluorescence assays (Fig. 4e).

**Podocytes**. Due to unique morphology and function, podocytes express many cell-type-specific genes. A total of 318 genes were highly expressed in both species, many of which are known to have an essential role in podocytes, such as *NPHS1*, *NPHS2* and *SYNPO* (Fig. 5a, b and Supplementary Data 9). However, we observed remarkably that 47 podocyte-expressed genes were specific to human and 33 to mouse podocytes (Fig. 5c). Independent bulk and scRNA-seq data validated this consistent differential expression pattern between the two species for most of the genes (Fig. 5d, e). As mouse glomeruli were isolated using the perfusion with beads, we also generated data using the bead-free glomerulus isolation method[28] (Supplementary Fig. 15). This approach gave similar results (Fig. 5e), indicating that the species diversity detected was independent of different isolation methods. Evolutionary *Nfasc* expression in the glomerulus was conserved in primates (human and cynomolgus monkey) but not detected in minipigs or rodents (rats and mouse) as shown by RT-PCR analysis of isolated renal fractions (Fig. 5f). The species-specific

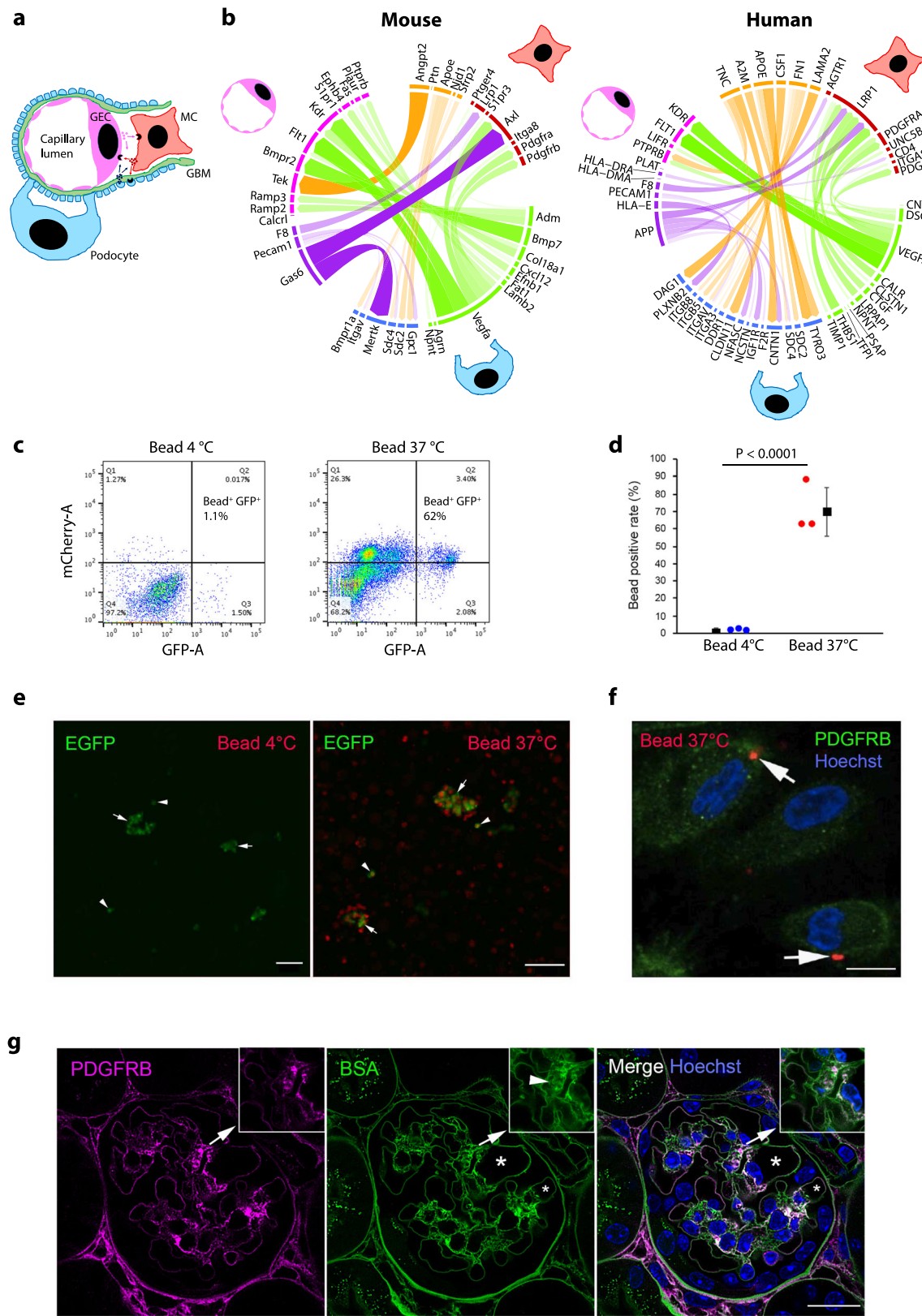

expression of this gene at the transcriptional level was further validated by Western blot analysis at the protein level (Fig. 5g).

**Glomerular endothelial cells**. *Ehd3*+ and *Kdr*+ endothelial cells represented GECs (Supplementary Fig. 3b–d). Thus, this group

was used for comparative analysis between two species. A total of 170 GEC genes showed high expression in both species, of which five genes (*Ehd3*, *Kdr*, *Igfbp5*, *Emcn* and *Tmem204*) were specifically upregulated in GECs compared to other captured renal cell types (Fig. 6a, b and Supplementary Data 10). In contrast to podocytes and MCs, we detected only six human-specific and no

**Fig. 3 Crosstalk between principal glomerular cell types and phagocytic activity in mesangial cells. a** Schematic relation of three principal glomerular cell types (MCs, GECs and podocytes). MC mesangial cell, GEC glomerular endothelial cell. **b** Chord diagrams showing ligand–receptor interactions between MCs, GECs and podocytes identified in mouse and human scRNA-seq datasets. Colours correspond to cell types and ligand/receptor categories, MCs (ligands: orange, receptors: dark red), GECs (ligands: purple, receptors: hot pink) and podocytes (ligands: lime, receptors: blue). The thickness and opacity of the arcs are proportional to the weights of ligand–receptor interactions as used in the NicheNet model. **c** Ex vivo phagocytosis by mouse glomerular cells. FACS scatter plots show gating of viable glomerular bead+ GFP+ cells isolated from *Pdgfrb*-EGFP mice. Conjugated pH-sensitive fluorescence is detectable in phagocytosed cells by the mCherry channel. Baseline was determined by incubating cells with beads at 4 °C (left). Notably, 62% of glomerular GFP+ cells are bead+ (phagocytic) when incubated with beads at 37 °C (right). Original flow cytometry plots and gating raw data for all three independent assays are available in the Source data file. **d** Dot plot showing percentage of bead+ cells in two groups. Percentages (%) of bead-positive cells in triplicate independent assays in 4 °C- and 37 °C-treated groups are presented as dots. Each assay included at least two mice. Error bars are defined as mean values and SD. The two-sided $P$ value of $6.5 \times 10^{-71}$ was calculated using the proportion test. **e** Visualization of phagocytosis by EGFP+ cells. In the left image, at 4 °C, fluorescent signal (red) of beads is not detectable. Arrows and arrowheads indicate partially digested glomerular EGFP+ MCs and EGFP+ single cells, respectively. In the right image, at 37 °C, partially digested glomerular EGFP+ MCs co-localize with fluorescent beads (arrows). Bead+ EGFP+ single cells are also visible (arrowheads). Scale bars: 50 μm. **f** In vitro phagocytosis by human glomerular PDGFRB+ cells. Fluorescent beads (arrows) are detected inside cultured PDGFRB+ (green) human glomerular cells. Hoechst-stained cellular nuclei. Scale bar: 100 μm. **g** Super-resolution STED microscopy analysis of mouse mesangial accumulation of injected FITC-BSA in vivo. Immunostaining of PDGFRB labelling MCs (violet) and FITC signal of BSA (green) in the mouse glomerulus 1 h after intravenous injection of FITC-BSA. Zoomed mesangium (insets) is indicated by arrows. FITC accumulation is indicated by arrowhead in inset. Glomerular capillary lumens are marked with *. Scale bar: 100 μm. Source data.

mouse-specific genes in GECs (Fig. 6c). Data from cells isolated using the bead-free method gave similar results indicating that the species differences detected were not due to divergent isolation methods (Fig. 6c). Moreover, validation using two independent scRNA-seq data[5,6], derived from mouse and human kidney, confirmed five of the genes (*IL13RA1*, *RXFP1*, *GIMAP7*, *PLA1A* and *APLNR*) to be human specific (Fig. 6d). Violin plots for *RXFP1*, as an example of a human-specific gene, demonstrated a robust expression in human GECs and absence in mouse (Fig. 6e). Evolutionarily, *RXFP1* expression in glomeruli was conserved in primates (human and cynomolgus monkey) but not detected in minipigs or rodents (rat, mouse) as shown by PCR analysis of isolated renal fractions (Fig. 6f).

**Identification of mouse PECs and macula densa cells.** In mouse, we identified a cluster of PECs highly expressing *Cldn1* (Fig. 7a, b), a classical marker for PECs[3] and *Pdgfrb* (Fig. 1b) These cells also express *Wt1*, a classical podocyte marker (Fig. 7a, b). As a TF, *Wt1* regulon was active in both PECs and podocytes, although its activation in PECs seemed to be heterogeneous in comparison with the podocytes (Fig. 7c), suggesting a potential genetic link between two cell types. A number of genes, including many PEC markers, were highly specific for PECs when compared to all annotated cell types, suggesting a unique function for this cell type (Fig. 7d). The molecular signature of PECs was further validated by data from *Pdgfrb*-EGFP mice, from which EGFP+ PECs were collected (Fig. 7e and Supplementary Fig. 10a). LBP and DKK3 were confirmed as PEC markers using immunostaining (Fig. 7f). We failed to identify human PECs in our scRNA-seq data, probably due to limited sample size.

Finally, we analysed the topological relations of captured tubular epithelial cells annotated as PTCs, collecting duct (CD), DCT and cortical thick ascending limb of the loop of Henle (cTAL) cells (Fig. 1b, Supplementary Fig. 16, and Supplementary Data 11, 12). Interestingly, re-clustering of cTAL and DCT2 cells, which were topologically close with each other, revealed a small cell population ($n = 11$) that was highly similar to cTAL cells, but topologically located between DCT and cTAL cells (Fig. 8a–c). They expressed classical macula densa cell (MDC) markers[29], such as *Slc9a2*, *Ptgs2*, *Slc12a1* and *Nos1* (Fig. 8d), but we could identify also several highly enriched genes in this population (Fig. 8e). To validate that the cells represented true MDCs, we performed double labelling experiments with the identified genes and a known marker, NOS1. In mouse kidney tissue, NT5C1A protein was clearly found in the same small subpopulation of

tubular cells (next to the glomerulus) as NOS1 (Fig. 8f). Similarly, SLC29A1 protein was detected in this group of cells in mouse kidney (Fig. 8f). These two candidates co-localized with NOS1 also in human kidney tissue (Fig. 8f). Finally, data from Human Protein Atlas (www.proteinatlas.org) supported our data as two different antibodies directed against NT5C1A gave strong positive signal in a subpopulation of tubular cells located close to the root of the glomerulus (Fig. 8g). Taken together, we identified MDCs in our scRNA-seq data and unravelled markers and biology for this rare cell population.

**Discussion**

scRNA-seq is a powerful technique to study individual cell types, particularly when cell-type-specific markers are unavailable. In the present study, we defined the precise cellular identity of the classical intraglomerular MC and a unique mural cell type located in the stalk/root of the glomerular tuft. The comparative analysis between human and mouse further identified sets of genes showing species-dependent differential expression profiles for each principal glomerular cell type. Our findings provide insights into the biology of glomerular mesangium and will be invaluable for translational studies.

MCs were proposed to be specialized microvascular pericytes by Schlöndorff already 30 years ago[30] and since then a lot of evidence has accumulated to support this. Like microvascular pericytes, MCs constitutively express PDGFRB and need the PDGFB–PDGFRB signalling axis to be recruited to the vascular unit[20]. PDGFRB has often been used as a marker for MCs, but it is also expressed in vSMCs and fibroblasts[20]. Although previous single-cell studies had annotated MCs, the heterogeneity within those MLCs was not explored and most markers used for defining MCs were not specific to true intraglomerular MCs[5,6]. Compared to previous kidney-related scRNA-seq publications[4–6], our study represents the first one using the Smart-seq2 platform. The relatively small number of FACS-sorted single cells with full-length read coverage contributes to higher gene detection rate and better data quality, which significantly improved downstream analyses[9]. This was especially helpful in clustering cell populations with high similarities, such as MLCs. Our scRNA-seq analysis demonstrated four subpopulations of glomerular-associated *Pdgfrb*+ cells, including bona fide intraglomerular MCs, vSMCs, renin-producing cells and a glomerulus-associated mural cell type located at the stalk region of the glomerulus tuft. It is clear that no single marker is exclusively expressed in individual MLC populations except for renin cells. However, we

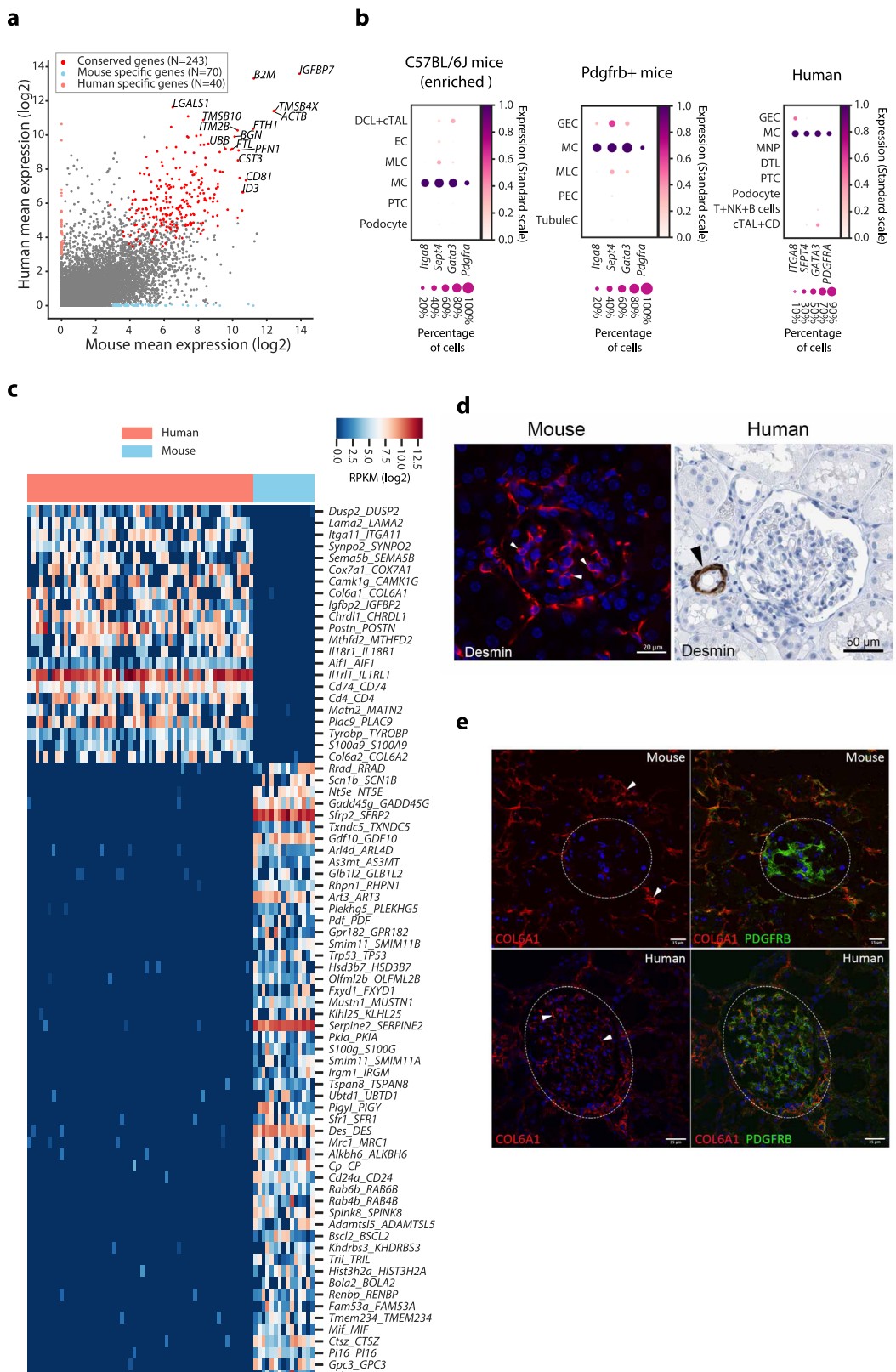

identified a set of genes (*Pdgfrb*, *Pdgfra*, *Gata3* and *Cnn1*) that can define these cellular identities within glomerulus-associated cells. It is crucial to use a right combination of molecular markers to identify bona fide MCs when for instance isolating cells for in vitro culture studies.

The definition of the MC molecular signature unravelled that they do not only possess a pericyte-like signature, but also a prominent fibroblast-like profile, characterized for instance by the expression of *Pdgfra*, *Dcn*, *Col12a1* and *Mmp2*. Pericyte-like properties are probably necessary to modulate the glomerulus

**Fig. 4 Differential and conserved gene expression profile of MCs between mouse and human. a** The expression (mean $\log_2$-transformed RPKM) of mouse and human gene homologues in MCs. Human- ($n = 40$) and mouse-specific ($n = 70$) genes are indicated in salmon and light blue, respectively. Conserved genes highly expressed in both species ($n = 243$) are indicated in red. Detailed information can be found in Supplementary Data 8. **b** The expression of conserved MC overexpressing genes in single-cell data generated from C57BL/6J mice with viable CD45$^-$ CD31$^-$ cell sorting (left), *Pdgfrb*-EGFP mice (middle) and human donor biopsies (right). The colour intensity and size of each dot represents the mean expression (standard scale) and the percentage of cells expressing each gene in individual cell types, respectively. **c** Heatmap showing the expression of differentially expressed genes in MCs between mouse and human. The colour scale is defined by $\log_2$(mean RPKM). **d** Validation of mouse-specific desmin expression in MCs. Immunofluorescence staining for desmin (red) shows abundant expression in the mouse glomerulus (arrowheads). Expression of desmin is undetectable in human glomeruli (www.proteinatlas.org, https://creativecommons.org/licenses/by-sa/3.0/), but strong signal in vSMCs (arrowheads) is detected as a positive control. Scale bars are indicated in images. **e** Validation of human-specific COL6A1 expression in MCs. COL6A1 (red) is detected in human MCs (arrowheads), whereas in mouse no significant reactivity in glomeruli is observed. PDGFRB (green) labels MCs in the glomerulus. Abundant extraglomerular interstitial reactivity for COL6A1 (arrowheads) serves as a positive control in mouse. Discontinuous circles indicate glomeruli. Scale bars: 15 μm for mouse and 35 μm for human.

filtration rate, whereas fibroblast-like functions are important to maintain the composition of mesangial matrix that 'glues' glomerular capillaries together and repair the mesangial damage[25]. In disease states, it is likely that fibroblast-like properties dominate with mesangial matrix expansion and fibrosis. Thus, we propose that MCs should be considered to be a unique glomerular cell type that is a hybrid of pericyte-vSMC-fibroblast.

We identified phagocytic machinery in MCs and validated its activity using in vitro and in vivo models. The presence of phagocytic function in MCs was proposed long time ago but this has been a controversial issue[2]. For instance, it has been suggested that only a proportion of MCs have this feature[31]. MCs are separated from the capillary lumen only by fenestrated GECs and matrix, and large molecules, such as albumin, can easily transverse these layers and reach MCs. Thus, it is plausible that MCs could have an important clearing role in the glomerular filtration barrier in which they can phagocytose entrapped macromolecules and prevent the filter from clogging. LRP1, a multifunctional scavenger receptor that can mediate the phagocytosis of many ligands[26], was highly expressed by MCs in our scRNA-seq data, and interestingly only lowly expressed by other MLCs. LRP1 is an obvious candidate to mediate phagocytosis, and further studies are warranted to explore the molecular mechanisms of phagocytosis in MCs.

Interestingly, we identified a unique mural cell population located at the glomerular stalk/root region that had a very distinct molecular profile from other cell types. Electron microscopic studies in 1960s described a cell group, referring to as Lacis or Goormaghtigh cells, located in JGA in continuity with MCs[22,32]. More recently, few reports have described cells that reside in the extracellular matrix between the glomerular tuft and the JGA that is contiguous to intraglomerular mesangium, and therefore named the cell group as extraglomerular MCs[33]. This cell population has been suggested a role in glomerular mesangial repopulation after injury through migration[33]. Our interpretation is that Lacis cells and EMCs represent the same cell population and correspond to the unique glomerulus-associated mural cell population identified in our study. An important finding in this study is that this cell population has a clearly different molecular profile from MCs, suggesting a different phenotype and function for these two cell populations. Therefore, we propose to re-designate them to their original name, the Lacis cell.

The expression switch from Lacis cells to MCs is remarkable, leading to the MC cluster far from the other three clusters in the UMAP space. The expression of smooth muscle genes (*Acta2/Tagln*) seems to correlate with a phenotypic transition from vSMCs to Lacis cells and then to MCs. This could explain previous descriptive observations regarding the expression switch of *Acta2* in MCs during glomerular development and in mesangial injury, where Lacis cells abundantly expressing *Acta2* possibly

migrate to the glomerulus and undergo further differentiation to mature MCs with downregulation of *Acta2*, *Tagln* and *Myh11* as well as upregulation of *Pdgfrb* and *Cspg4* under the glomerular microvascular environment. Of note, we observed Lacis cells specifically expressing *Ephx3* and *Olfr78*, which have been shown to play a role in regulating blood pressure[34,35]. This suggests that Lacis cells act not only as a reserve replenishing MCs, but may also contribute to other unknown functions. All these suggest the presence of two cellular entities with distinct functions: MCs in the glomerulus and Lacis cells within the glomerular central stalk and JGA. Further studies on origin and biology of Lacis cells are obviously needed.

Mouse is the most commonly used animal to model renal glomerular diseases. The field is struggling with poor translation between mouse and man[13,15]. In this study, we generate the most detailed scRNA-seq profiles of three principal glomerular cell types in two species, and demonstrate major species diversities that can explain the lack of translation. Importantly, we could validate this expressional evolutionary shift by analysing human, cynomolgus monkey, minipig, rat and mouse kidney tissues, further highlighting the remarkable differences in glomerular molecular signatures. Obviously, our data will help to design and interpret translational studies. An example is the observed discrepancy in RXFP1 expression between primate and rodent kidney. We previously identified RXFP1 as one of the most highly enriched glomerular GPCRs when compared to the tubulointerstitial fraction of human kidney[36]. The current study is the first to describe RXFP1 expression in the GECs of the human kidney and its absence in any cell type of the murine/rat/pig glomerulus. This glomerular RXFP1 expression is particularly noteworthy considering that relaxin (serelaxin), the native ligand to this receptor that has been evaluated in clinical trials, exerts specific renal vasodilating activity and potentially reno-pretection[37]. Our data indicate that neither rodents nor pigs are suitable to study kidney-protective effects of relaxin. Moreover, our data validate recent findings that mouse is not a suitable model to study MN caused by circulating PLA2R1 antibodies as mouse podocytes (in contrast to human) do not express PLA2R1.

PECs are unique epithelial cells surrounding Bowman's space and have been suggested to act as stem cells for both podocytes and PTCs[3]. Moreover, they seem to have an important role in the pathogenesis of many glomerulopathies via activation/proliferation to contribute to crescent formation. In this study, we provide a global transcript profile of mouse PECs, which will be useful for understanding the basic biology of these cells, as well as provide tools to isolate them. We detected similarity with podocytes, suggesting a link between the two cell types.

Highly rare MDCs play a key role in regulating glomerular blood filtration via the tubuloglomerular feedback mechanism[29]. Using Smart-seq2, we identified and profiled three cell types

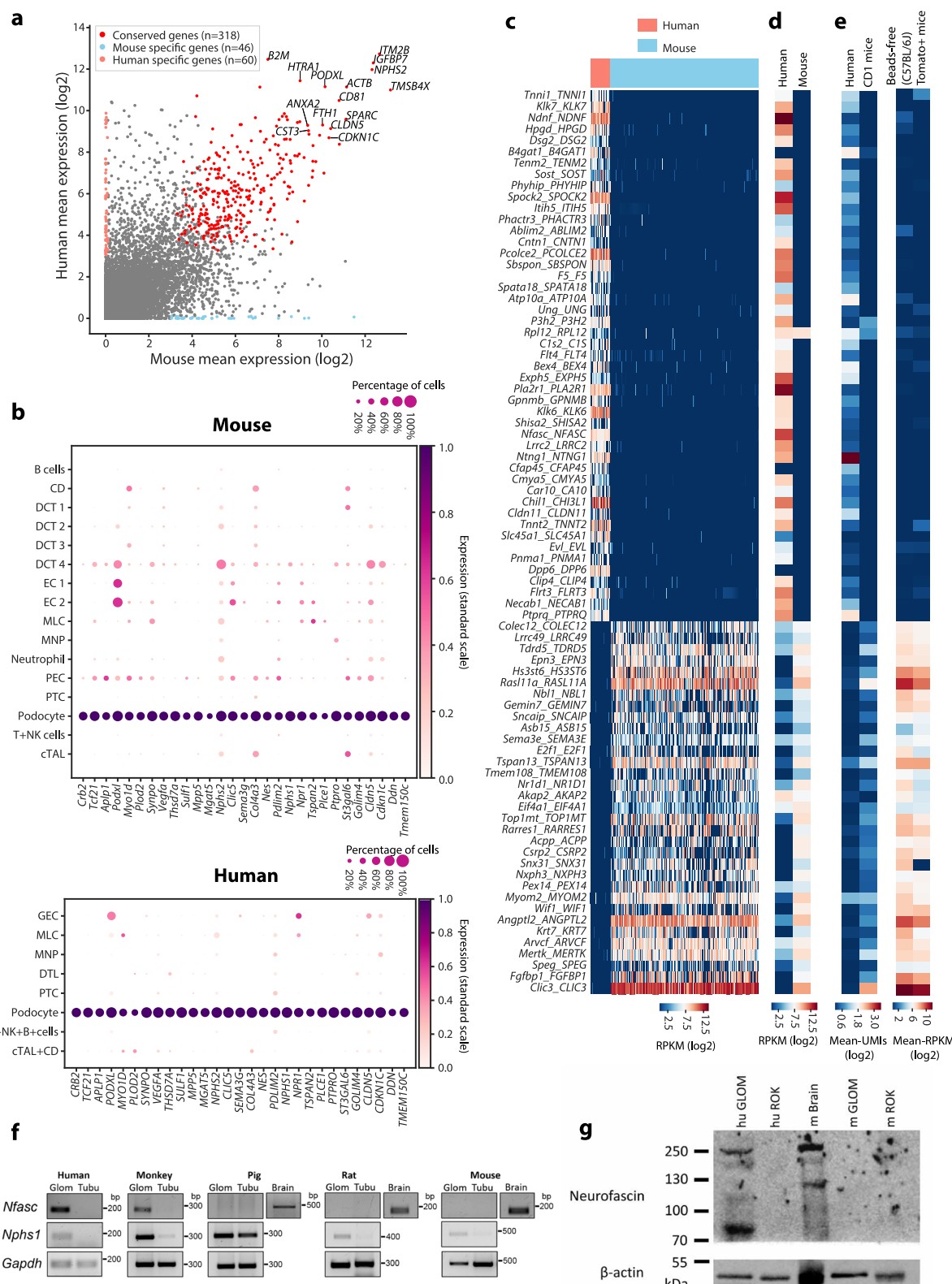

(MDCs, EMCs and renin-producing cells) that form the JGA. Interestingly, we identified in MDCs a molecular pair: a cytosolic nucleotidase NT5C1A catalysing the conversion of AMP to adenosine, and a corresponding adenosine transporter SLC29A1. As adenosine is known to be important for the function of JGA, we propose that this pair is likely to be involved in the adenosine-mediated tubuloglomerular feedback mechanism. Subsequent

scRNA-seq studies, especially in disease states, are warranted to unravel the complex crosstalk between JGA cells.

In summary, based on single-cell transcriptome data from mouse and human, we defined the precise cellular identity of two glomerulus-associated mural cell types and highlight the species diversity of molecular signatures in principal glomerular cell types. The results provide insights into the role of mesangium in

**Fig. 5 Differential and conserved podocyte gene expression profile between mouse and human. a** The expression (mean $\log_2$-transformed RPKM) of mouse and human gene homologues in podocytes. Human- ($n = 60$) and mouse-specific ($n = 46$) genes are indicated in salmon and light blue, respectively. Conserved genes highly expressed in both species ($n = 318$) are indicated in red. Detailed information can be found in Supplementary Data 9. **b** The expression of conserved podocyte highly expressing genes in identified cell types in mouse (C57BL/6J) (upper panel) and human (lower panel). The colour intensity and size of each dot represents the mean expression (standard scale) and the percentage of cells expressing each gene in individual cell types, respectively. **c** Heatmap showing the levels of differentially expressed genes in podocytes between mouse and human. The colour scale is defined by $\log_2$(mean RPKM). **d** In silico validation of podocyte species-specific genes using published bulk podocyte RNA-seq data from mouse and human. The expression levels are shown as $\log_2$-transformed RPKM. **e** In silico validation of podocyte species-specific genes using published (left) and in-house (right) scRNA-seq data of annotated human and mouse podocytes. Expression levels are shown as $\log_2$-scale average UMI counts (published data) and as $\log_2$-scale average RPKM (in-house data). **f** Experimental validation for the gene expression of human-specific gene *NFASC*. RT-PCR was used to detect the expression of *Nfasc* in kidney fractions (glom: glomerulus, tubu: tubule) isolated from human, cynomolgus monkey, minipig, rat and mouse. *Nphs1* and *Gapdh* were used as loading controls. cDNA generated from the brain tissue served as an experimental positive control. The molecular-weight size of PCR amplicons is indicated in the right side as bp. **g** Western blot analysis of NFASC in mouse and human glomeruli. Lysates of human and mouse glomeruli (hu GLOM, m GLOM) and tubules (hu ROK, m ROK, glomerulus-free or rest of kidney) were used. As a positive control, the mouse brain lysate (m Brain) was used. β-actin was used as a loading control. Protein molecular mass (kDa) is shown on the left side of the blot image. Of note, the blots for NFACS and β-actin were derived from the same Western blot, but were treated separately for the exposure processes due to notably weaker signal for NFASC than β-actin. The original raw data for blots and PCR gel images are available in the Source data file. Source data

renal biology and chronic kidney diseases, as well as are essential to design and interpret translational studies.

## Methods

**Study design**. The objective of this study was to define the precise cellular identity of glomerular cell types in mouse and human at the single-cell resolution and further detect species diversity of glomerular cell transcriptomes. To achieve the aim of the study, we set up three key items in study design: (1) Smart-seq2 as the platform of scRNA-seq to generate deeper data per cell in comparison to previous studies; (2) kidney samples from healthy living donor kidney biopsies, and wt adult male C57BL/6J mice; (3) single cells from isolated glomeruli. scRNA-seq data generated by independent cell sorting, immunostaining, together with functional assays, were used to validate the findings at the protein level and functionality. To exclude potential false positive in comparative analysis for species diversity, stringent criteria were used followed by validations using multiple independent datasets.

**Ethical considerations**. The study design and conduct complied with all relevant regulations regarding the use of human study participants and was conducted in accordance to the criteria set by the Declaration of Helsinki. All participants provided written informed consent. The use of human material for studies was approved by the local Ethics Review Authority (www.etikprovningsmyndigheten. se) in Stockholm, Sweden, archive numbers 2010/579-31 and 2016/615-32. All methods used in human material were carried out in accordance with relevant guidelines and regulations defined in the ethical permit.

For mouse work, all experimental protocols were approved by The Linköping Ethical Committee for Research Animals ("Linköpings djurförsöksetiska nämnd"), Linköping, Sweden (archive number DNR 41-15). All methods used in mouse experiments were carried out in accordance with relevant guidelines and regulations defined in the ethical permit. All animals were housed in standard, single ventilated cages with 12 h light–12 h dark cycle, and had ad libitum access to water and chow. The house temperature was maintained as $20 \pm 2\,°C$ and the relative humidity was kept as $50 \pm 5\%$.

**Kidneys**. For mouse kidneys, totally 16 wt adult male C57BL/6J mice (mean age of $12.6 \pm 2.6$ weeks old) were used. Among them, ten mice were used for unbiased glomerular single-cell sorting and six mice for double CD45[−] and CD31[−] cell sorting. As an independent validation data, two male *Pdgfrb*-EGFP[+/−] reporter mice (12 weeks old) were used via sorting EGFP[+] cells. In addition, tdTomato-positive podocytes from two male mice, generated by crossing the Pod-Cre female mice with the floxed STOP tdTomato male mice, were sorted for scRNA-seq as a validation dataset.

For human kidneys, we collected normal kidney biopsies from eight living healthy kidney donors (three male and five female, $48.2 \pm 12$ years old). During kidney transplantation surgery, donor kidneys were nephrectomised and the biopsies were taken within 3 min after shutting off blood circulation in the kidney. Clinical records verified that all these living donors were healthy, including exclusion of hypertension, diabetes and any other disorders possibly affecting kidney function.

**Isolation of glomeruli**. To isolate mouse glomeruli, anesthetised mice first were perfused with 1× HBSS to remove circulating blood, followed by perfusion of Dynabead (14013, Thermo Fisher). Minced kidneys were digested with collagenase

IV (1 mg/ml, Thermo Fisher) plus DNase I (50 U/ml, Thermo Fisher) at 37 °C for 30 min. After passing through two 100-μm strainers, cell suspensions were pelleted by centrifugation ($500 \times g$ at 4 °C for 5 min). The glomerular pellet was re-suspended in HBSS and then the beads-containing glomeruli were gathered by the magnet holder and washed three times.

To isolate human glomeruli (mean 150 μm in diameter), we considered maximally to include Bowman's capsule, which is often stripped off during classical serial sieving (425-, 250- and 150-μm sieves). To this end, we used only a single 300-μm sieve allowing easy passage of human glomeruli together with Bowman's capsule. Briefly, minced kidney biopsies were first incubated with collagenase IV (1 mg/ml) plus DNase I (50 U/ml) at 37 °C for 15 min, followed by a single sieving using a 300-μm strainer (PluriStrainer). Then a 100-μm strainer (Pluristrainer) was used to separate human glomeruli from tubules. Glomeruli retained on 100-μm strainers were collected by turning the strainer upside down followed by multiple washes and pelleted by centrifugation ($500 \times g$ at 4 °C for 5 min). Purity of isolated glomeruli was validated under a dissection microscope. All kidney tissues were kept on ice except for during enzymatic digestion.

To exclude whether bead perfusion and the magnet separation process in glomerulus isolation could lead to subsequent transcriptome alterations compared with the sieving method, we isolated mouse glomeruli using a recently reported bead-free method without bead perfusion[28]. In brief, minced mouse kidneys were incubated with collagenase IV (1 mg/ml) without DNase in 1× HBSS buffer at 37 °C for 15 min. After spinning down, the re-suspend tissue mixture passed through two sieves from 100- to 75-μm strainers. The filtrate containing glomeruli and tubular fragments was collected using a 40-μm strainer. Then enriched glomeruli were obtained after settling the mixture on a 10-cm culture dish for 1–2 min allowing adherence of tubules to the dish bottom. We unbiasedly sorted these cells.

**Single-cell dissociation**. The purified glomeruli were digested by incubation in PBS containing Ca[++]/Mg[++], collagenase IV (1 mg/ml), pronase (1 mg/ml, Sigma-Aldrich) and DNase I (50 U/ml) at 37 °C for 15 min with shaking at 400 r.p.m. For mouse glomeruli, we added an additional 15-min digestion to further dissociate undigested glomeruli. Dissociated single cells were pelleted by centrifugation ($450 \times g$ at 4 °C for 5 min). Single cells were washed once with a sorting buffer (PBS without Ca[++]/Mg[++], 1% FCS and 1 mM EDTA) and then passed through 50-μm filters (Filcon cup-type, BD Biosciences).

**FACS sorting**. BD Aria series or a FACS Melody sorter (BD Biosciences) equipped with a 384-well plate stage was used for single-cell sorting. Two sorting strategies (unbiased and biased) were performed in this study. First, to sort all cell types, we stained viable cells using CellTracker, CMFDA-Green (1:1000, Thermo Fisher), and then unbiasedly sorted dye-positive cells to 384-well plates (Supplementary Fig. 17a). In general, glomerular cells from an adult mouse were sufficient to sort 1–2 plates and cells from each human kidney biopsy consisting of 10–15 glomeruli were able to sort 250–380 cells. Second, to sort certain cell groups, a biased method was also performed. For instance, to enrich MLCs, CD45[+] immune cells and CD31[+] endothelial cells were stained and then were excluded during sorting. Briefly, glomerular cells from six wt mice were pooled and incubated in 50 μl of sorting buffer containing mouse anti-bodies CD45-PE-CF594 and CD31-APC on ice for 15 min followed by washing. Cells positive for the live cell dye and negative for CD45 and CD31 were gated for sorting (Supplementary Fig. 17b). By this design, glomerular cells could completely exclude *Cd45*[+] immune cells, but we still found *Pecam1*[+] endothelial cells after scRNA-seq, probably due to enzymatic destruction of CD31 epitopes. To sort EGFP[+] cells isolated from two *Pdgfrb*-EGFP mice, we gated cells positive for DRAQ5 (1:2000, Thermo

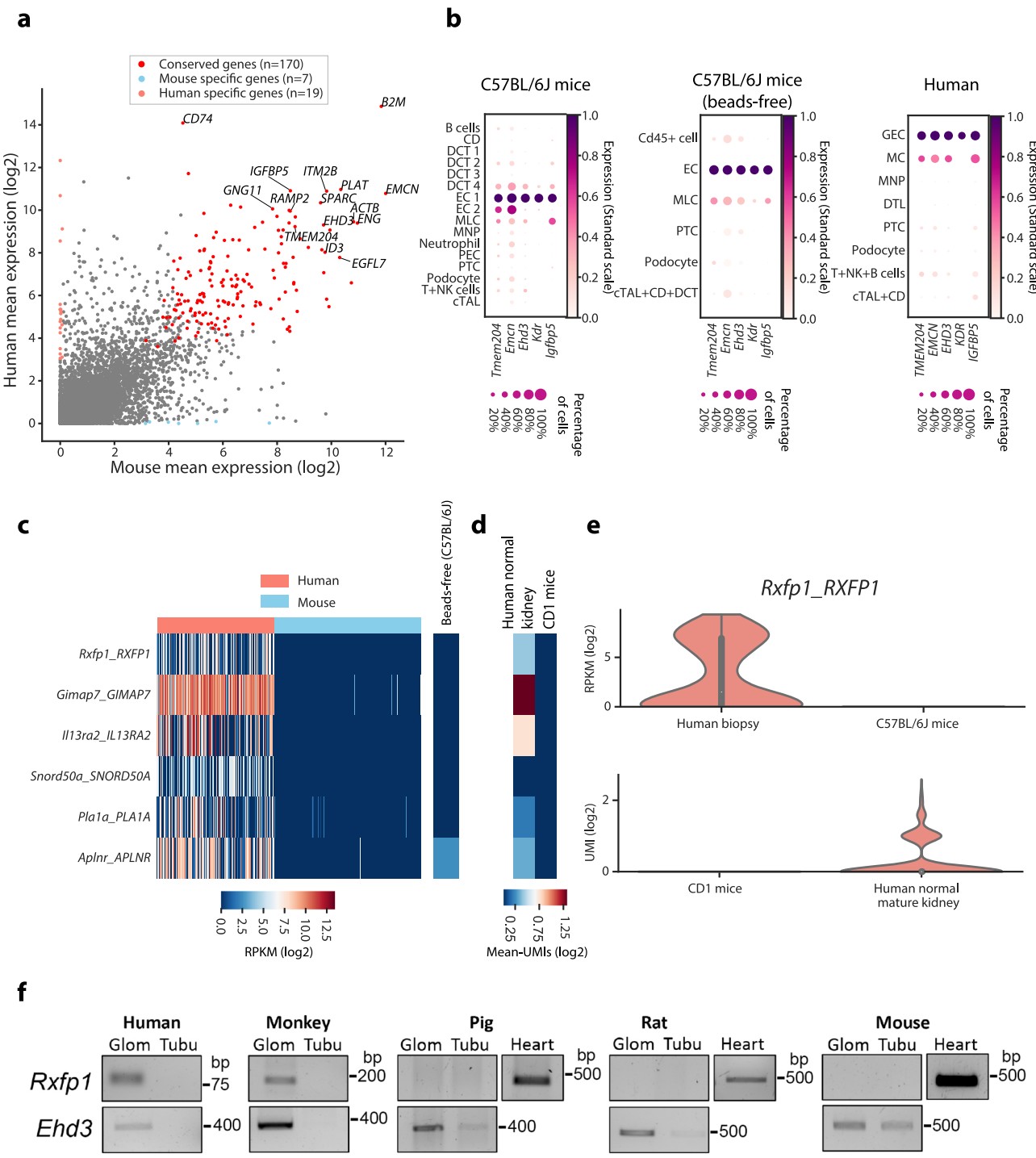

fisher) and EGFP for sorting. As a control for autofluorescence, kidney cells from wt mice were used (Supplementary Fig. 17c). Similarly, we sorted podocytes positive for both tdTomato and CMFDA-Green. The flow cytometry analysis software BD FACSDiva v8.0.2 installed in the BD Aria or FACSChorus v1.1 in the BD FACS Melody sorter was used for data analysis as well as single-cell sorting.

**Smart-seq2 protocol**. We generated full-length cDNA by precisely following the Smart-seq2 protocol[38]. Briefly, mRNA in cell lysis of 384-well plates receiving sorted single cells was reversely transcribed into cDNA using oligo(dT) primer and SuperScript II reverse transcriptase (Thermo Fisher). The second strand cDNA was synthesized using a template switching oligo. The process was run at 42 °C for 90 min. The synthesized cDNA was then amplified for 23 cycles by PCR. After bead-assisted purification, cDNA quality was analysed by using 2100 Bioanalyzer with a DNA High Sensitive chip (Agilent Biotechnologies). If the cDNA passed the quality control, cDNA was tagmented using Tn5 transposase followed by a standard index PCR using Illumina Nextera XT index kits (set A-D). The sequencing was performed on HISEQ 3000 (Illumina). Sequencing of all sorted plates was performed by the single-cell core facility at ICMC, Karolinska Institute.

**scRNA-seq data processing, quality control and filtering**. Raw sequencing data were de-multiplexed and converted into fastq format for each cell by using Illumina bcl2fastq with default setting. Reads from mouse and human data were aligned to mouse genome mm10 and human genome hg38, respectively, using the STAR aligner[39]. Uniquely aligned reads mapping to the RefSeq gene annotations were used for gene expression estimates at reads per kilobase transcript and million mapped reads (RPKMs) using rpkmforgenes[40]. Low-quality cells were excluded from downstream analysis when they failed to meet the following criteria for retaining cells: (1) ≥50,000 sequence reads; (2) ≥40% of reads uniquely aligned to the genome; (3) ≥40% of these reads mapping to RefSeq annotated exons; (4) <10% of uniquely mapped reads from ERCC spike-ins; and (5) ≥500 genes with RPKM ≥1. In addition, doublets detected by Scrublet[41] were further removed. A total of 2277 out of 3938 mouse and 766 out of 2188 human single cells from unbiased

**Fig. 6 Differential and conserved glomerular endothelial cell gene expression profile between mouse and human. a** The expression (mean log$_2$-transformed RPKM) of mouse and human gene homologues in GECs. Human- (*n* = 19) and mouse-specific (*n* = 7) genes are indicated in salmon and light blue, respectively. Conserved genes highly expressed in both species (*n* = 170) are indicated in red. Detailed information is found in Supplementary Data 10. **b** The expression of conserved GEC highly expressing genes in identified cell types in data generated from C57BL/6J mice (left panel), C57BL/6J mice where glomeruli were isolated using a bead-free glomerulus isolation method (middle) and human donor biopsies (right panel). The colour intensity and size of each dot represents the mean expression (standard scale) and the percentage of cells expressing each gene in individual cell types, respectively. **c** Heatmap showing the expression of differentially expressed genes in GECs between mouse and human. Only six human-specific genes and no mouse-specific genes were detected in GECs (left). These human-specific genes were not expressed or expressed at very low level in our in-house C57BL/6J mice data generated using the bead-free isolation method (right). The colour scale is defined by log$_2$(mean RPKM). **d** In silico validation of GEC species-specific genes using published scRNA-seq data. Expression levels are shown as log$_2$-scale average UMI counts. **e** Violin plots showing the expression of *RXFP1* in GECs of human (*n* = 241 cells) and mouse (*n* = 305 cells) from this study and in GECs from two independent data (*n* = 1556 mouse cells and *n* = 731 human cells). The miniature box plot in the violin plot illustrates the first quartile, median and the third quartile with whiskers of maximum 1.5 IQR (the interquartile range). **f** Experimental validation for the gene expression of human-specific gene *RXFP1*. RT-PCR was used to detect expression of *Rxfp1* in kidney fractions (glom: glomerulus, tubu: tubule) isolated from human, cynomolgus monkey, minipig, rat and mouse. *Ehd3* was used as loading controls. cDNA generated from heart tissues served as an experimental positive control. The molecular-weight size of PCR amplicons is indicated in the right side as bp. The original PCR gel images are available in the Source data file. Source data

sorting, 600 out of 1152 mouse cells by CD45$^-$ CD31$^-$ cell sorting, 344 out of 384 EGFP$^+$ cells from *Pdgfrb*-EGFP mice, 163 out of 192 cells from mouse glomeruli isolated by using the bead-free method and 182 out of 205 mouse tdTomato$^+$ podocytes passed the quality control and were included in downstream analyses.

**Single-cell cell type assignments**. We first identified genes with true biological variability (FDR < 0.01) from technical noise using a quantitative statistical method[42] and used the resulting genes for PCA dimensionality reduction. To identify cell clusters, the top significant principle components determined by 1000 random permutations based on the JackStraw approach[43] were used for affinity propagation clustering analysis. The single cells were then projected onto a two-dimensional UMAP space for visualization. Each cell cluster was assigned to a corresponding cell type based on known cell type marker genes, for example, *Nphs1* for podocytes, *Pecam1* for endothelial cells, *Pdgfrb* for MLCs, *Cldn1* for PECs, *Pck1*, *Aqp2* and *Umod* for tubular segment cells and *Ptprc* for leucocytes.

**Differential expression analysis between defined groups**. To identify genes differentially expressed between defined cell/sample groups, we performed Kruskal–Wallis test with Benjamini–Hochberg multiple testing correction on the log$_2$-transformed expression data. For differential expression analysis between more than two groups, post-hoc tests were performed on all possible pairwise group comparisons. Genes that were significantly upregulated in one group vs the other groups (adjusted *P* < 0.01 and minimum fold change of 2) were considered as group-specific upregulated genes.

**Aligning mouse MLC data from two different experimental batches**. For in-depth study of a heterogeneity within mouse MLCs, we merged all mouse MLCs (*n* = 339) obtained from two independent sources by unbiased (Supplementary Fig. 1a) and biased sorting using anti-CD45 and CD31 antibodies (Supplementary Fig. 6a). The CD45$^-$ CD31$^-$ cell data showed higher gene detection rate than the unbiased sorted data (Supplementary Fig. 4a, b). The batch effects between the two mouse MLC datasets were adjusted using the empirical Bayes method ComBat[44].

**Phenotype prediction of MLCs using cell scoring of probabilistic similarity**. Cell scoring of probabilistic similarity of each cell type relative to the defined cell types at the transcriptional level was used to predict phenotypes of MLC populations. Log$_2$-transformed data were scaled by Minmax normalization, and then applied for learning approach of L2 regularized logistic regression[45,46]. In this classifier model, we trained the model to learn the general prototypes of defined lung vascular and perivascular cell types[22], including pericytes, vSMCs, vascular fibroblast-like cells and various endothelial cell types (arterial EC, capillary EC, EC1, EC2). To train the model, we obtained the overdispersed genes by estimating the mean and coefficients of variation, and then calculated the enrichment score. The overdispersed genes were further ranked by two heuristics for cell-type specificity of both fold-change and enrichment score-change[47]. Thus, the ranked marker genes of defined cell-types were used for the learning model. To choose the adequate regularization strength, the classifier accuracy and sum of regression coefficients were inspected against regularization strength. The classifier accuracy was estimated by a *k*-fold cross-validation, of which the dataset was randomly split for 35 iterations (25% test_size). The value of regularization strength (0.02) was chosen corresponding to the maximum point of learning curve reaching the accuracy plateaus. The ready learning model was used to predict the probabilities of each cell belonging to each trained reference cell-types by using the soft-argmax algorithm. The permutation test of dataset was applied to qualify the significance of the prediction. Data were visualized on box-swarm plot and Radar plot[48]. Box-swarm plot represents the probabilities of each cell assigning to the reference cell-

types, and each dot represents the probability of one single cell assigning into this cell-type. Radar plot consists of a sequence of equiangular polygon spokes with the distal vertex representing each trained reference cell-types. Here, the polygon spokes were removed to make the plot clear. The distance between the polygon centre and each vertex of the polygon represents the relative probabilities of each trained reference cell assigning to the defined reference cell-types. Thus, the position of each predicting cell was calculated as a linear combination of the probabilities against all reference cell types, and then visualized as the relative position to all vertices of the polygon.

**Mapping the immune cells to human kidney immune cell atlas**. Totally 7803 immune cells from the published immune cell atlas of mature human kidney from a patient undergoing tumour nephrectomy[17] were collected and used as a reference atlas for kidney-associated immune cell types. These cells were aligned together with all captured mouse and human immune cells by unbiased sorting using Seurat v3 method[49]. In addition, the probabilistic cell scores were calculated for each cell to predict the similarities to the reference immune cell types (Supplementary Fig. 2b). The expression of immune cell type markers was visualized in UMAP including all aligned cells (Supplementary Fig. 2c).

**Trajectory analysis of MLC subpopulations and cells within different tubular zonations**. To reconstruct the biological relations of identified MLC subpopulations, we generated topology-preserving maps of single cells using the partition-based graph abstraction method (PAGA)[50]. It resulted in better cell group-level relations than noisy relations among individual cells with high transcriptomic similarities (Figs. 2a and 8b). In addition to PAGA, diffusion map and pseudotime cell trajectories[51] were performed to reveal a global structure of zonation dynamics within mouse tubular cells (Fig. 8c and Supplementary Fig. 16a).

**Identification of conserved and species-specific genes between mouse and human**. The mouse and human gene homologues were obtained from Ensembl database (v92) using BioMart (http://apr2018.archive.ensembl.org/biomart/martview/). Genes with multiple orthologues or with no expression detected in both species were excluded from the analysis. For each principal glomerular cell type, genes with the median log$_2$-transformed RPKM ≥4 in both species were considered as conserved high expression genes. The criteria for significant expressional difference between species in each cell type were: (1) mean log$_2$-transformed RPKM < 0.1 in one species and mean log$_2$-RPKM ≥3 in another species; (2) significant for the Kruskal–Wallis test with Benjamini–Hochberg multiple testing correction (adjusted *P* < 0.01); and (3) a species-specific gene does not simultaneously exist in multiple glomerular cell types. For MCs, only defined mouse glomerular MCs captured by CD45$^-$ CD31$^-$ cell sorting were compared with human MCs due to the fact that a few of MCs were captured by unbiased cell sorting. For podocytes and GECs, the comparisons were based on the data from unbiased cell sorting. To validate analysis data, three sources RNA-seq datasets were included: (1) two bulk RNA-seq datasets from sorted mouse[52] and human podocytes[53]; (2) two scRNA-seq datasets from mouse and human with cell-type annotation of *Nphs1$^+$* podocytes and *Pecam1$^+$* and *Ehd3$^+$* GECs[5,6]; and (3) our own C57BL/6J mouse scRNA-seq data using the bead-free isolation method and unpublished scRNA-seq data from sorted mouse tdTomato$^+$ podocytes.

**TF network and crosstalk analysis**. The activity score of each TF regulatory module was calculated using the SCENIC method[54]. Binary regulon activity matrix was used for UMAP and heatmap visualizations (Figs. 1d, 2i, 7b and Supplementary Fig. 5). For ligand–receptor crosstalk analysis, we focused on the interactions among the three principal glomerular cell types (GEC, MC, podocyte) using

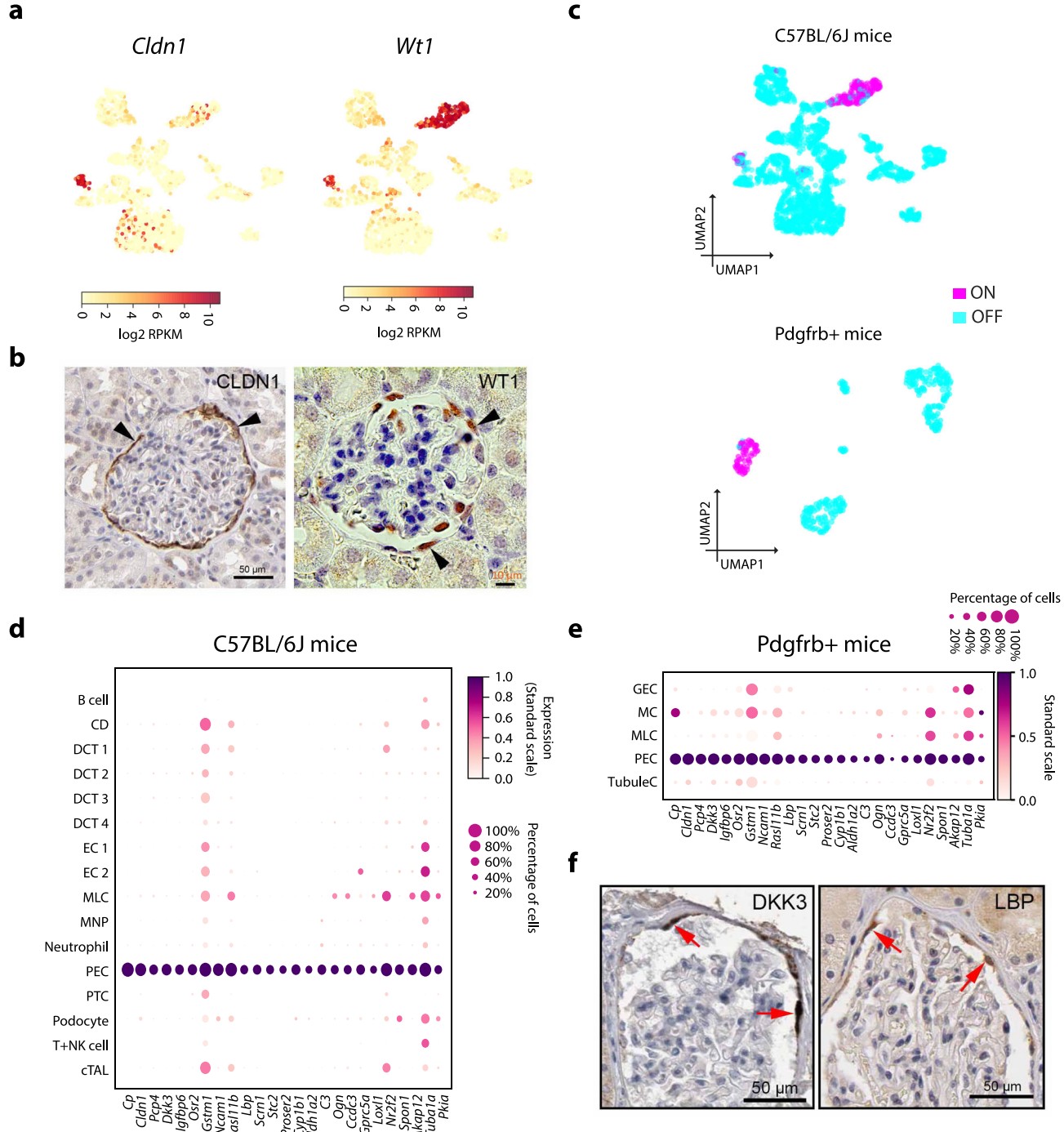

**Fig. 7 Identification of mouse glomerular parietal epithelial cells (PECs). a** The expression of *Cldn1* and *Wt1* in the mouse glomerulus. *Wt1* is abundantly expressed in mouse *Nphs1*-expressing podocytes and the *Cldn1*-expressing cluster. The colour scale is defined by $\log_2$(mean RPKM). **b** Immunohistochemistry confirms the localization of human CLDN1 and mouse WT1 in PECs on Bowman's capsule (arrowheads). The CLDN1 image is downloaded from www.proteinatlas.org (https://creativecommons.org/licenses/by-sa/3.0/). **c** UMAP showing the *Wt1* regulon activity in single cells of C57BL/6J mice (upper) and Pdgfrb⁺ mice (bottom). **d** The expression of 25 genes significantly overexpressed in PECs compared to all other identified cell types from C57BL/6J mice. The colour intensity and size of each dot represent the mean expression (standard scale) and the percentage of cells expressing each gene (*x*-axis) in individual cell types (*y*-axis), respectively. **e** Validation of the 25 PEC signatures in an independent data from *Pdgfrb*-EGFP mice. **f** Immunohistochemical staining of DKK3 and LBP in human Bowman's capsule. Staining for DKK3 and LBP in PECs are indicated (red arrows). The images are downloaded from www.proteinatlas.org (https://creativecommons.org/licenses/by-sa/3.0/). Scale bars: 50 μm.

one cell type as receiver cells and the other two cell types as sender cells. Differential expression analysis was first performed to retrieve cell-type-specific signatures, followed by selection of confident ligand–receptor pairs for the signatures using NicheNet[55]. The candidates of ligand–receptor interactions were then visualized in Chord diagram grouped by each cell type with distinct colour representing ligands and dark colour representing receptors as described in figure legends.

**Immunostaining and Western blot.** Immunofluorescence staining of cryosection from mouse or human was performed according to standard procedures. In brief, kidney tissues were embedded in OCT compound and frozen on dry ice. Frozen blocks were sectioned using a CryoStat NX70 (Thermo Fisher) to 5–8 μm-thick sections. The sections were stored at −80 °C. For frozen section staining, tissues were thawed at room temperature (RT) for 30 min and then fixed in cold acetone at

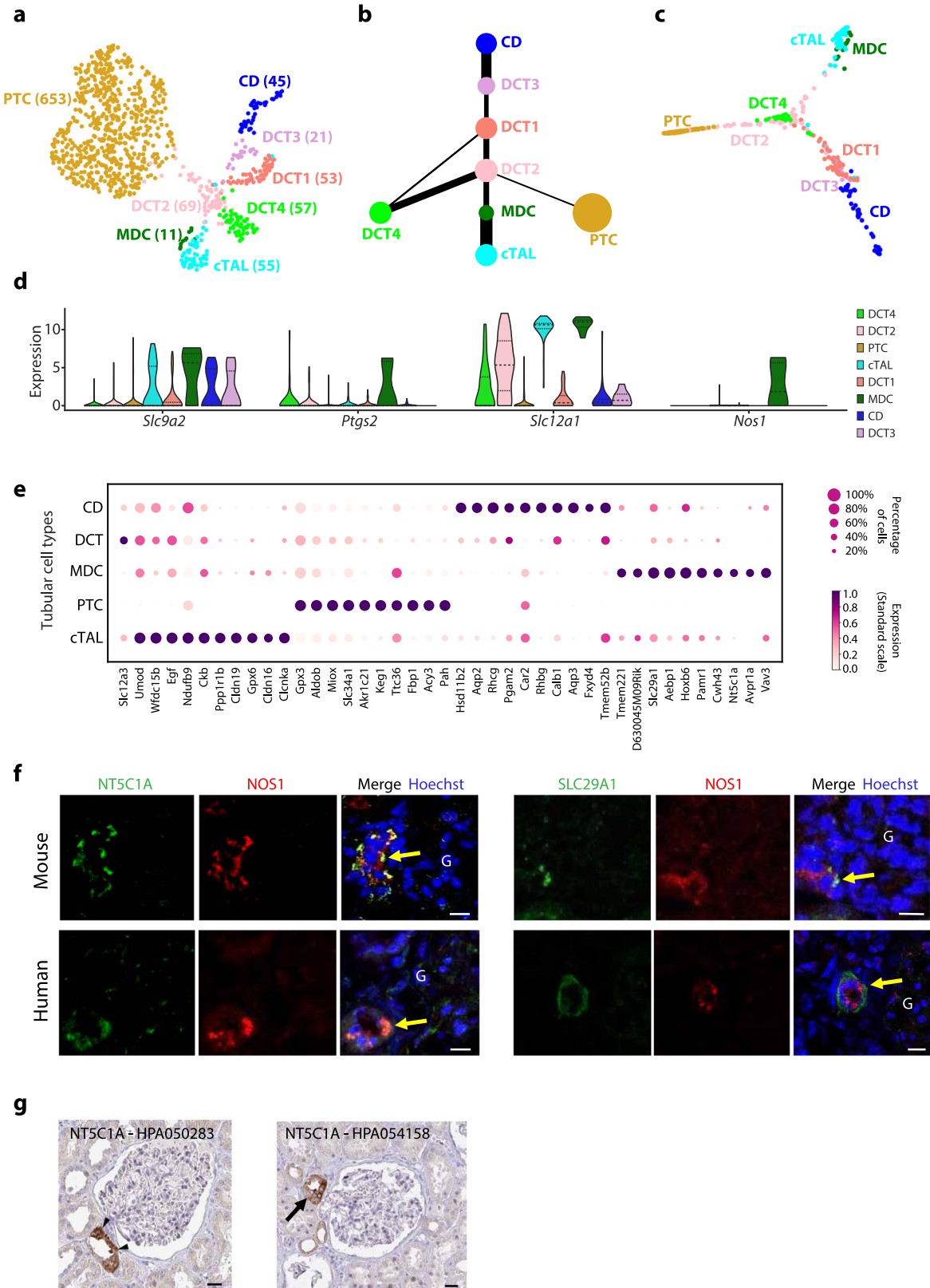

−20 °C for 10 min. After washing, tissues were blocked for 1 h at RT. Thereafter, sections were sequentially incubated with primary antibodies at 37 °C for 1 h or 4 °C overnight, and corresponding conjugated secondary antibodies at 37 °C for 1 h. Nuclei were stained using Hoechst 33342 (1:10,000 Thermo Fisher) at RT for 5 min. For immunochemistry staining of paraffin-embedded kidney sections, kidney tissues were fixed in 4% paraformaldehyde for 1 day and embedded in paraffin following the standard protocol. Paraffin-embedded tissues were sectioned to 6-μm

thickness. After deparaffinisation, antigen retrieval was done by heating at 98 °C for 20 min. The primary antibody was incubated at 4 °C overnight. Then we used a Vectastain Elite ABC HRP kit (PK-7200 Vector Labs) for the subsequent staining procedure and followed the manufacture's protocol. Nuclear staining was performed using hematoxylin. The confocal microscope Leica TCS SP8 or Zeiss SLM 710 was used for fluorescence imaging. We also used immunohistochemistry images of Human Protein Atlas (www.proteinatlas.org, https://creativecommons.

**Fig. 8 Identification of macula densa cells from captured mouse tubular cells. a** UMAP of captured tubular cells from C57BL/6J mice coloured by cell types and DCT subclusters. CD collecting duct, DCT distal convoluted tubule, PTC proximal tubule cell, MDC macula densa cell, cTAL cortical thick ascending limb. Colour dots indicate diverse subsets. **b** PAGA graph of cell type topological relations. **c** Two-dimensional diffusion map trajectory of tubular cells. **d** The expression (log$_2$-transformed RPKM) of classical MDC markers in tubular cell populations. Colours represent tubular corresponding subsets described in **a**. **e** The expression of top 10 genes significantly overexpressed in each identified tubular cell subpopulation. **f** Double immunofluorescence staining for NT5C1A (green) and SLC29A1 (green) with NOS1 (red) in mouse (left) and human kidney tissues (right). Both proteins NT5C1A and SLC29A1 localize to the same subpopulation of NOS1-positive tubular cells (yellow arrows) next to the glomerulus (G). **g** Immunostaining for NT5C1A using two different antibodies (HPA HPA050283—left and HPA054158—right) show strong reactivity in a small subpopulation of tubular cells next to the root of the glomerular tuft (arrowheads or arrow). The images are downloaded from www.proteinatlas.org (https://creativecommons.org/licenses/by-sa/3.0/). Scale bars (**f**, **g**): 30 μm.

org/licenses/by-sa/3.0/) for cellular localization of the proteins of interest. Western blotting was performed according to the standard protocol. Briefly, isolated mouse and human glomeruli were obtained by using the bead-free glomerulus isolation method and whole glomerular tissues were processed using RIPA buffer containing complete protease inhibitors (Roche). Denatured protein lysates were separated using NuPAGE 4–12% Bis-Tris Protein gels (Invitrogen) and transferred to nitrocellulose membranes. Incubation with primary antibodies and appropriate secondary antibodies was performed to detect the proteins. Information of all primary and secondary antibodies used in this study is presented in Supplementary Table 1. For immunochemical staining of WT1, a biotinylated anti-rabbit secondary antibody in the Vectastain Elite ABC HRP kit (PK-7200 Vector Labs) was used.

**RT-PCR.** Human glomeruli were isolated from nephrectomy kidneys and other animal glomeruli were isolated from wt mouse, rat, minipig and cynomolgus monkey kidneys, respectively. Total RNA was purified from freshly isolated glomeruli using the RNeasy Mini kit (Qiagen). The first-strand cDNA synthesis was carried out using iScript cDNA synthesis kit (Bio-Rad). Mouse and rat multiple tissue cDNA panels (#636745 and #636751, TakaraBio) were used as positive controls. PCR was carried out according to standard procedures. All primer sequences used in this study are presented in Supplementary Table 2. To be comparable among PCR products of diverse species, Tm of primers was designed to be similar. The PCR reaction in 20 μl containing 200 μM dNTP, 1.5 mM MgCl$_2$, 0.25 μM primers, 10–20 ng cDNA and 1.25 U HotStart Taq DNA polymerase (Qiagen) was performed onto a PCR cycler (Bio Rad) with a starting denature of 95 °C for 15 min, followed by 28 reaction cycles (95 °C for 30 s, 60 °C for 45 s and 72 °C for 1 min). The size of the PCR amplicons was determined by running 1% agarose gel electrophoresis with GeneRuler 1 Kb plus DNA ladder (Thermo Fisher).

**Ex vivo phagocytosis assay in mouse.** Due to lack of conjugated antibodies to specifically label enzymatically dissociated MCs, we chose EGFP$^+$ glomerular cells from *Pdgfrb*-EGFP reporter mice to estimate MCs. Protonex Red 600-Latex Bead Conjugate (AAT Bioquest, 21209), in which latex beads with mean size of 0.7 μm are conjugated with a pH-sensitive fluorescence, was used for phagocytosis assay analysed by FACS. Freshly isolated glomerular single cells from *Pdgfrb*-EGFP mice ($n = 7$) using the bead-assisted method described above were suspended in 100 μl of 1× HBSS containing 20 mM HEPES, and 5 μl of fully mixed latex beads then was added, followed by incubation at 37 °C for 30 min with mild shaking. Phagocytosis was terminated by adding cold HBSS buffer and cells were washed twice. To determine the baseline of bead-positive cells, wt mouse kidney tubular cells were incubated with 5 μl of beads at 4 °C for 30 min, where phagocytic activity is silent and the conjugated fluorescence maintains inactivation according to the product instruction. In FACS analysis, we detected DRAQ5-labelled viable cells (APC-Cy7 channel), EGFP (GFP channel) and bead fluorescence Red 600 (mCherry channel). Their autofluorescence is shown in Supplementary Fig. 17d. To gate valid bead$^+$ cells that phagocytosed beads, we first gated singlet and live cells. Based on the baseline of bead$^+$ cells at 4 °C and wt mouse cells for autofluorescent EGFP, gating thresholds of EGFP and mCherry were determined. Flow cytometry data analysis was performed using FACSChorus v1.1 in the BD FACS Melody or FlowJo v10. To visualize bead$^+$ EGFP$^+$ MCs, we isolated glomerular cells from *Pdgfrb*-EGFP mice and treated them with beads at 4 °C and 37 °C, respectively. Concentrated cell suspension (100 μl) was added onto a 35-mm glass-bottom dish (MatTek.com) for confocal microscopic imaging. The Zeiss confocal microscope (SLM 710) was used to detect co-localization of GFP$^+$ cells with mCherry$^+$ beads.

**In vitro phagocytosis assay in human glomerular cells.** Glomeruli were isolated using a standard sieving method from the tumour-free pole of kidneys that were nephrectomised due to renal cancer. Glomerular tufts were resuspended with a cell culture media (RPMI 1640 media containing 10% FBS, 1% penicillin/streptomycin and 1% insulin transferring selenium A; all products were purchased from Thermo Fisher). Glass coverslips were coated with fibronectin (Sigma-Aldrich) at the concentration of 3 μg/cm$^2$ according to the manufacturer's instructions. The glomeruli were seeded onto the coated coverslips and maintained at 33 °C and 5% of

CO$_2$ in cell culture media. On day 14, phagocytosis assay was performed using Protonex Red 600-Latex Bead Conjugate (AAT Bioquest, 21209) according to manufacturer's instructions. Then, cells were fixed with 4% PFA-PBS and stained with anti-mouse PDGFRB primary antibody (MAB1263, 1:1000, R&D Systems), followed by an Alexa fluor 488 anti-mouse secondary antibody (A-11001, 1:3000, Thermo Fisher). Imaging was performed with Leica SP8 confocal microscope.

**Uptake of albumin by MCs in vivo.** Wt mice ($n = 3$) were intraperitoneally injected either with 400 μl of 0.5 mg/ml BSA or BSA-fluorescein isothiocyanate conjugate (Merck KgaA, Darmstadt, Germany). After 1 h, the mice were sacrificed, and kidneys were fixed in 10% neutral buffered formalin solution at 4 °C overnight. After this, the kidney pieces were cut into 0.3-mm-thick slices using a Vibratome. During immunolabelling, PBST (PBS 1× with 0.1% vol/vol Triton-X) was used as a diluent in all steps. Samples were incubated in anti-Pdgfrb antibody for 24 h at 37 °C, washed in PBST for 10 min at 37 °C followed by Alexa fluor 647 anti-mouse secondary antibody (A-31571, 1:200, Thermo Fisher) incubation for 24 h at 37 °C and washed for 10 min at 37 °C prior to mounting. DAPI (D1306, Thermo Fisher, 1:1000) was added together with the secondary antibody to visualize nuclei. Prior to imaging, samples were immersed in 80.2% (w/w) fructose with 0.25% 1-thioglycerol and mounted in a MatTek dish (MatTek P35G-1.5-14-C). A Leica SP8 3X STED system was used for imaging.

**Statistics and reproducibility.** The Chi-square test with Yates correction or Fisher's exact test was used to estimate statistical significance of mouse phagocytosed cells between bead$^+$ and bead$^-$ groups and two-sided $P$ values of 0.05 were chosen as the significant threshold. In addition, the proportion test was used for $P$ value calculation in Fig. 3d and Supplementary Fig. 13b. All mouse experiments were repeated at least three times using biologically independent replicates. All antibody immunostaining experiments were performed at least two times using identical or different antibody combinations in analysing kidney tissues from at least three individual mice or humans.

**Reporting summary.** Further information on research design is available in the Nature Research Reporting Summary linked to this article.

## Data availability
Raw sequencing data of mouse Smart-seq2 scRNA-seq together with processed data of human and mouse used in this study have been deposited in Gene Expression Omnibus (GEO) with the accession number of GSE160048. Due to EU General Data Protection Regulation (GDPR), raw sequencing data of human Smart-seq2 scRNA-seq have been submitted to European Genome-Phenome Archive (EGA) with the accession number EGAD00001006861. The access to raw human sequencing data is granted upon contacting the Data Access Committee (for details: https://ega-archive.org/access/data-access). In addition, a public searchable database of our kidney glomerular single-cell atlas can be accessed via https://patrakkalab.se/kidney. Source data are provided with this paper.

## Code availability
All the computational analyses were performed using Python and R programming languages. Scripts of key steps can be found at https://github.com/PingChen-Angela/scRNA_kidney_NatureCommunications2021.

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

## Acknowledgements

We acknowledge the single-cell core facility at ICMC for scRNA-seq work and contribution of Drs. Jing Guo and Patricia Rodriguez for initial protocol tests of kidney single-cell preparation and Dr. Aleksandra Krstic at Clinical Pathology and Cytology, Huddinge University Hospital, for help with FACS cell sorting. We also thank the Advanced Light Microscopy Facility at SciLifeLab, Solna, Sweden, for support with acquisition of microscopy images. This project was financially supported by AstraZeneca, Swedish Diabetes Foundation, Swedish Kidney Foundation, Westmans Foundation and Karolinska Institute Foundation and Grants from Center for Innovative Medicine.

## Author contributions

B.H., P.C. and J.P. conceived and designed the project, B.H. established the methods for single-cell preparation, performed mouse and human cell preparation, cell sorting and FACS analysis of phagocytosis assay, P.C. performed all computational analysis and prepared the figures, S.Z., D.D. and E.C. performed the mouse perfusion and mouse cell preparation, S.Z. performed the immunostainings and Western blots, D.D. performed the RT-PCRs, D.U-J, B.S. and T.B. were responsible for super-resolution imaging, G.G.K. performed the in vitro phagocytosis assay, K.M-H. performed the in vivo phagocytosis assay, Y.H. performed the data analysis of cell scoring, M.J. and M.B-M. performed the immunostainings and confocal imaging, A.Wi., L.W. and A.We. coordinated the collection of human kidney biopsies, P.E. conceived the data analysis of cell scoring, C.B.

conceived the project and coordinated RNA sequencing, M.L. participated in data analysis and study design, R.S. supervised the data analysis, B.H., P.C. and J.P. wrote the paper with the assistance from the other authors.

## Funding

## Competing interests
AstraZeneca is financially supporting C.B., R.S. and J.P. laboratories, M.L. is an employee of AstraZeneca, C.B. is a consultant for AstraZeneca. The remaining authors declare no competing interests.
