## [Peer Review File · Nature Communications]

Reviewers' comments:

Reviewer #1 (Remarks to the Author):

Single cell technologies are increasingly used to study the composition and cellular interactions within tissues. This study aims to understand the cellular composition of isolated mouse and human glomeruli.

The design of the study, looking for unbiased single cell data and comparing to CD45/CD31 FACS sorted cells from the same tissue, as well as cells obtained from reporter mouse model is, on principle, a good design to address the fundamental questions about cellular composition of the glomerulus (especially given that podocytes, for example, or PECs, are indeed rare cells). However, several technical and biological considerations significantly limit enthusiasm for this study.

First and foremost, the number of cells sequenced in this study is many orders of magnitude less than several other studies already published in this field including by Suzstak et al., Behjati et al., Clatworthy et al., Kretzler et al, and several others who have collectively sequenced more than 400,000 human and mouse kidney cells including cells from isolated glomeruli (as here). The low number of cells severely limits the statistical power of this study. Signals not detected may be missing because of low numbers of cells sequenced, and signals obtained may be misrepresenting the underlying tissue being studied, again, due to lack of statistical power. This is a major flaw in this study, meaning, the only way to improve this work would be to repeat the experiment with several orders of magnitude more cells in order to draw conclusions that lend confidence.

Another technical point: FACS-based smart seq2 technology has now been significantly overtaken by droplet-based technologies and in fact this may be why the authors were limited in the number of cells sequenced, but this unfortunately is a serious technical and statistical limitation for this study.

On the subject that smart-Seq affords deeper sequencing of each cell, this would make sense to state as an advantage if the study had indeed identified a novel pathway or receptor-ligand interaction not previously known, afforded to the authors by deeper sequencing. As it stands, this is not the case, so smart-seq2 sequencing has not resulted in new insights in this study.

Some specific comments:

Figures 1 and 2: all the markers identified in both mouse and human have been previously described by several other studies, so there is no new insights or conclusions drawn by these analyses.

Figures 4-6: the comparison of mouse and human podocytes, MCs and GECs is interesting, but quite similar to observations made by Harder et al. and others as above.

The identification of Laci cells as extraglomerular mesangial cells (also known as Polkissen cells, or Goormaghtigh cells) is interesting as a confirmation of past work showing the same.

Reviewer #2 (Remarks to the Author):

He et al describe an elegant scRNA-seq study and reveal insights into the composition and heterogeneity of mesangial cells and some potential interspecies differences in glomerular gene expression patterns. Some concerns reduce enthusiasm for the manuscript. For example, the isolation methods likely introduce some differences in cellular composition between human and mouse samples. In addition, the numbers of cells derived (~800 from 8 human donors, ~1600 from 16 mice) seems low and raises concern for inefficient isolation and subsequent biased cell sampling/representation. Conversely, the number of genes recovered per cell is high compared to

other scRNA-seq studies, increasing confidence in the differential expression patterns described by the authors. I think the strongest points of the manuscript are a) heterogeneity of mouse mesangial like cells with identification of a genomic fingerprint of Lacis cells and b) interspecies differences in podocytes. Additional protein-level validation of the key findings would greatly strengthen the authors' conclusions. This is especially necessary in the absence of specific mechanistic insights.

Specific comments:

- More information is required about how the 8 human donor kidney biopsies were collected. Were these patients undergoing a procedure (E.g. nephrectomy)? Was history of diabetes/hypertension an exclusion criterion? What was the age of the donors? Is the human data pooled from all 8 individuals (i.e. only ~100 cells/individual). In general, the numbers of cells available for analysis should be clearly stated at the outset in the results section.

Perhaps because so few cells were isolated, the authors do not appear to have identified populations of immune cell types such as dendritic cells or resident macrophages in their cell clusters despite a strong literature that such cells are present in the kidney.

f

- The glomerular isolation methods are likely driving some of the cellular compositional differences between the human and mouse datasets. Serial sieving in human glomeruli usually strips off Bowman's capsule and therefore the lack of PECs is not surprising. Conversely, Dynabead assisted glomerular isolation in mice often retains Bowman's capsule thereby including PECs in the sample. This should be incorporated either into the methods or results section. More generally -- how much of the interspecies heterogeneity can be attributed to this difference in cellular sampling inherent to isolation?

- Fig. 2a is superfluous. Fig. 2b should include Wt1 to compare with podocytes as the other major cell type expressing this marker. Indeed, the assertion that WT1 is a highly specific marker of parietal epithelial cells, expressed in line 124, is erroneous as it is often used as a preferential marker of podocytes. It is thought the expression of this marker in parietal cells may represent a transitional stage by which parietal cells could differentiate into podocytes in certain disease and regenerative states. In Fig. 2c, rather than just showing staining for known markers of PECs, what about validating some of the novel markers identified by scRNA-seq? For the IHC panel, a zoom in on a single glomerulus with capsule would be easier to interpret the staining pattern at the cellular level

- Fig. 3 could benefit from localization of MLC1-4 using a panel of markers identified from the sequencing studies. The authors localize MLC-C1 (Lacis) and MLC-C4 (intraglomerular mesangial cells). Where do MLC-C2 and C3 cells localize in/around the glomerulus?

- I would caution against expansive statements such as "we here identified for the first time pure transcriptional profiles of MCs". From Fig. 3f, the true mesangial cell (MLC-C4) dataset appears to be derived from *only* 17 cells - and that too isolated from multiple mice. Also, the range of the Fig. 3f's heatmap could be reduced (e.g. from -2 to 4) to show more of the data since there appear to be very cells with expression values above 4.

- In Fig. 4a/5a/6a, highlighting some of the outlier and shared genes by callouts would be nice to see

- In Figures 4 and 6, validating Nfasc and Rxfp1 expression at the protein level would provide additional confirmation

I think the authors overstate the novelty of their finding that PLA2R is expressed by human podocytes and not those of mouse. This has been well recognized for some time by those who have been investigating the role of this protein in the pathogenesis of membranous nephropathy.

- The colors of the heatmaps vary across the figures. I understand that different measures are

being represented, but some common ones (e.g. log₂ RPKM) are shown in different color coordinates which is a bit jarring.

- A data sharing statement is not included in the manuscript. A portal to explore the data is now becoming standard for these types of studies and facilitates access to non-bioinformatically enabled researchers

Reviewer #3 (Remarks to the Author):

General comments

The authors present a dataset of murine and human single-cell RNA sequencing data from kidney glomerular cell preparations using the Smart-Seq2 technology. This has allowed them to generate a high quality dataset with sensitive gene detection, which complements recently published broader and larger scale droplet based single-cell and single-nucleus RNA sequencing approaches of the mouse and human kidney 1–4. The authors report a fairly comprehensive inventory of glomerular cell types and their transcriptional profiles, and are able to recover expected cell types albeit with the conspicuous absence of glomerular parietal epithelial cells in the human data. In the paper's current form, the work describes a spatially resolved inventory of glomerular cells, but does not reveal and substantially novel biology which would merit publication in Nature Communications in its current form.

The paper currently lacks one or more features which would significantly increase the impact of the paper such that it would be suitable for publication in Nature Communications:

Full "round trip" validation of identified cell populations - The paper is presented as a resource defining the "cellular identity of MCs". Whilst the authors show the expression of specific markers nominated from their scRNAseq data with microscopy, they then stop there. One approach would be to isolate cells on the basis of these markers using FACS, perform further scRNAseq, and illustrate that they recover the same transcriptionally distinct subsets with high fidelity. This approach has previously been employed in scRNAseq studies taking a similar focussed approach⁵. Characterisation of functional properties of identified populations - as described extensively in the introduction, the glomerulus is "functionally and structurally unique" arrangement of cells. The specialised function of this apparatus relies on interactions and signalling between cell types. Mesangial cells for example, have been ascribed numerous functions, for example: phagocytosis; roles as pericytes, extracellular matrix production; and for "Lacis cells" signalling in tubuloglomerular feedback. It would be interesting to build a richer picture of the interactions and functions of these cells, initially building from analysis of the scRNAseq data, but ultimately with some validation and functional assays. Which should build on hypotheses generated from investigation of the data. For example, do the mesangial cell populations express phagocytic machinery? If so, experiments to validate this function would be valuable.

Delineating conserved transcriptional programs at play between murine and human glomerular cell subsets in glomerular disease settings - Throughout the introduction and discussion the authors highlight the translational significance of understanding the molecular composition of glomerular cell types. Ideally the authors would extend their study by adding such translational significance. Kidneys from murine glomerular disease (eg diabetic injury, nephrotoxic nephritis etc.) could be compared to biopsies from kidneys with cognate human disease to uncover conserved glomerular cell type specific transcriptional patterns in disease - for example activation state associated with mesangial expansion in diabetic glomerular injury.

Specific comments

Number of cells: Lines 40-42: "To this end, we used Smart-seq2 to profile large numbers of individual glomerular cells isolated from human living donor renal biopsies and mouse kidney" - the authors should clarify the number of cells they analysed. How many cells failed quality control? Each legend should clearly state the number of cells analysed in each UMAP plot.

Reference for glomerular disease prevalence: Line 81-82: "Glomerular disease processes are a major clinical challenge as they account for 85% of cases leading to end-stage renal disease" requires a reference. Presumably the authors are happy to include secondary glomerular injury as a result diabetes mellitus and hypertension rather than only primary glomerulonephritides in their quoted prevalence estimate.

Previous report of murine mesangial cells: The authors should cite a study from 2017 which performed scRNAseq on murine mesangial cells using the Fluidigm C1 platform⁶.

Human glomerular parietal epithelial cells: As stated above, the absence of these cells in the human data is a clear weakness of the study, and should be addressed by generating additional human data to capture these cells before publication. This will allow the authors to extend their report on species conservation and differences between glomerular cells to glomerular parietal epithelial cells. Adding additional human data will substantially increase the value of this dataset as a resource.

Unclassified & unknown cells:

Fig 1B+C both show populations of unclassified cells. Fig S2A suggests that these are cells with low number of genes detected. Are these cells unclassifiable because they are of low quality? The authors should therefore consider removing these from the analysis or reviewing their quality control pipeline given that unclassifiable, low quality cells have made their way into the final dataset.

These plots also show clusters of cells which are annotated as "unknown". In contrast to the "unclassified" clusters, these appear to have a large number of genes detected. The authors should attempt to annotate these cells, at least to a broad lineage.

Murine data structure: An initial murine scRNAseq dataset is presented in Fig 1A, however on lines 147-149, the authors write: "To address this, we first increased the number of Pdgfrb+ cells by sequencing cells from 6 additional mice (Fig S3a-b) and then pooled and reclassified all Pdgfrb+ MLCs.." This implies that the murine data is fragmented into two datasets. The authors would have improved power to detect distinct cell populations in their murine data if they combined these data in the initial analysis (Fig 1B).

Quantitative assessment of similarity to other datasets: The authors make claims of similarity or dissimilarity to recently published datasets which are well supported by quantitative/statistical analyses - for example Fig 3G+H.

Immune cell marker genes:

Supplementary figure 1E+F uses the following genes to distinguish immune cell subsets: CD68, CD79A, CD79B, CD3D, CD3G, KLRK1, KLRD1, CSF3R. Can the authors clarify why they selected these genes in particular? Whilst CD3D and CD3G are good markers for T cells, CD68 is a tissue macrophage marker and not typically used as a monocyte marker, CSF3R can be expressed on granulocytes and monocytes and so is not a sensible choice for distinguishing these specifically. Suggested sets of genes commonly used for distinguishing these cell subsets are: CD4 T cells, CD3D, CD3G, TRAC, CD4; CD8 T cells, CD3D, CD3G, TRAC, CD8A, NKG7; B cells, MS4A1, HLA-DRA, CD19, HLA-DRA; monocytes, CD14 (classical), FCGR3A (non-classical), FCN1, VCAN, LYZ; neutrophils, FCGR3B, S100P, VCAN, NK cells; PRF1, NKG7, FCGR3A, NCAM1, KIR family genes. The authors may wish to utilise a recently published scRNAseq analysis of immune cells in the kidney to aid their classification of these cells², making qualitative or quantitative reference to this dataset.

The authors should display the single cell data rather than just boxplots.

Juxtaglomerular apparatus: It is surprising that the authors were unable to recover macula densa tubular cells given that they isolated glomeruli. Can the authors comment on why they think these cells were not recovered? Are they in fact within the populations shown in Fig1A+B but not correctly annotated?

Collagen type 4: Type 4 collagen form an essential component of the glomerular basement membrane and is affected in Alport disease and anti-GBM (Goodpasture's) disease. This study permits a unique opportunity to understand which cells produce this important feature of the filtration apparatus.

Can the authors comment on the expression patterns of collagen type 4 genes?

Can the authors clarify further why they instead chose to focus on COL6A1?

Description of mesangial cell phenotype: Lines 194-195: "...suggesting that MCs possess as complex phenotype that is a combination of pericytes, vSMCs and fibroblasts". The authors should tone down this statement and rephrase along these lines: "suggesting that MCs possess a complex phenotype with features of pericyte, vSMC, and fibroblast transcriptional programmes".

Transcription factor networks: There would be substantial value added to the paper through attempting to understand the transcription factor networks defining each cell type. This might be straightforwardly accomplished using the Pyscenic set of computational tools⁷ -

<https://github.com/aertslab/pySCENIC#id5>. This would help to address important questions this study leaves unanswered such as:

Given PEC and podocytes express WT1, are WT1 transcriptional targets active in both cell types? The authors argue that mesangial cells exhibit a phenotype that is a “combination of pericytes, vSMCs and fibroblasts” - is this determined by combinations of transcription factors targeting the same genes as in these cells?

Intercellular communication networks: It would be revealing to understand the putative cell-cell interaction networks between glomerular cell types. This could be accomplished using tools such as CellPhoneDB 8 or NicheNet9.

Gene names: Gene names should be italicised, and the authors should check the manuscript to ensure that they are using the correct capitalisation for murine vs human gene names, one such example is line 223 - “Violin plots of *Nfasc*, as an example of a human-specific gene, demonstrated a robust expression in human podocytes and its absence in mouse”.

Mesangial cell relationships - can the authors clarify further the relationship between the intra- and extra-glomerular mesangial cells they find. There is some effort made at implying a trajectory between these cells (Fig3A), so can this be extended by some determining some directionality (for example using RNA velocity10)? It would be very interesting if these cells interconvert.

Data availability: The data must be made available in raw and processed annotated form, ideally via a user friendly portal. This will maximise the utility of this data for the kidney research community. We would encourage the authors to submit their data to the Human Cell Atlas data coordination platform. There is no comment on data availability in the manuscript.

Code availability: The code used in the analysis should be published on github or similar as is standard in the field. There is no comment on code availability in the manuscript.

Supplementary data format: Supplementary data files should be provided as .csv not .xls files. There are well reported issues with corruption of gene names in .xls files 11.

References

1. Park, J. et al. Comprehensive single cell RNAseq analysis of the kidney reveals novel cell types and unexpected cell plasticity. doi:10.1101/203125.
2. Stewart, B. J. et al. Spatiotemporal immune zonation of the human kidney. *Science* 365, 1461–1466 (2019).
3. Young, M. D. et al. Single-cell transcriptomes from human kidneys reveal the cellular identity of renal tumors. *Science* 361, 594–599 (2018).
4. Lake, B. B. et al. A single-nucleus RNA-sequencing pipeline to decipher the molecular anatomy and pathophysiology of human kidneys. *Nat. Commun.* 10, 2832 (2019).
5. Villani, A.-C. et al. Single-cell RNA-seq reveals new types of human blood dendritic cells, monocytes, and progenitors. *Science* 356, (2017).
6. Lu, Y., Ye, Y., Yang, Q. & Shi, S. Single-cell RNA-sequence analysis of mouse glomerular mesangial cells uncovers mesangial cell essential genes. *Kidney Int.* 92, 504–513 (2017).
7. Aibar, S. et al. SCENIC: single-cell regulatory network inference and clustering. *Nat. Methods* 14, 1083–1086 (2017).
8. Efremova, M., Vento-Tormo, M., Teichmann, S. A. & Vento-Tormo, R. CellPhoneDB v2.0: Inferring cell-cell communication from combined expression of multi-subunit receptor-ligand complexes. doi:10.1101/680926.
9. Browaeys, R., Saelens, W. & Saeys, Y. NicheNet: modeling intercellular communication by linking ligands to target genes. *Nat. Methods* (2019) doi:10.1038/s41592-019-0667-5.
10. La Manno, G. et al. RNA velocity of single cells. *Nature* 560, 494–498 (2018).
11. Ziemann, M., Eren, Y. & El-Osta, A. Gene name errors are widespread in the scientific literature. *Genome Biol.* 17, 177 (2016).

Reviewers' comments:

Reviewer #1 (Remarks to the Author):

Single cell technologies are increasingly used to study the composition and cellular interactions within tissues. This study aims to understand the cellular composition of isolated mouse and human glomeruli.

The design of the study, looking for unbiased single cell data and comparing to CD45/CD31 FACS sorted cells from the same tissue, as well as cells obtained from reporter mouse model is, on principle, a good design to address the fundamental questions about cellular composition of the glomerulus (especially given that podocytes, for example, or PECs, are indeed rare cells). However, several technical and biological considerations significantly limit enthusiasm for this study.

First and foremost, the number of cells sequenced in this study is many orders of magnitude less than several other studies already published in this field including by Suzstak et al (Sci 2018/mu); Behjati et al (Sci 2018/hu), Clatworthy et al (Sci 2019/hu), Kretzler et al (Dev 2018/fetal hu), and several others who have collectively sequenced more than 400,000 human and mouse kidney cells including cells from isolated glomeruli (as here). The low number of cells severely limits the statistical power of this study. Signals not detected may be missing because of low numbers of cells sequenced, and signals obtained may be misrepresenting the underlying tissue being studied, again, due to lack of statistical power. This is a major flaw in this study, meaning, the only way to improve this work would be to repeat the experiment with several orders of magnitude more cells in order to draw conclusions that lend confidence.

Another technical point: FACS-based smart seq2 technology has now been significantly overtaken by droplet-based technologies and in fact this may be why the authors were limited in the number of cells sequenced, but this unfortunately is a serious technical and statistical limitation for this study.

We disagree partially with this comment. We argue that it is more important to have deep (i.e. a large fraction of the cells' transcriptomes covered) and clean (i.e. non-contaminated) data from relatively fewer cells than to cover large numbers of cells that are contaminated or that do not include all important cell types or that small number of genes are detected. For instance, the first published scRNA-seq study of kidney tissue (Park et al, Science 2018; cited by reviewer as an example of what we should aspire to) covered almost 58 000 renal cells. In spite of this, however, they failed to capture intra- and extra-glomerular mesangial cells, parietal epithelial cells of the glomerulus and macula densa cells. Outside of the glomerulus they also failed to capture renal fibroblasts (although they claimed to have done so). The lack of depth in that and other studies have hampered the identification of different renal subpopulations. Our Smart-seq2 approach generated clearly deeper data from each cell in comparison to previous studies in renal tissue (Fig 1A below).

We used Smart-seq2 platform, which is particularly suitable for the analysis of rare cell types, or any cells that are hard to capture in sufficient numbers using conventional microfluidic-based methods. Therefore, our choice of methodology is more feasible to identify and define rare cell populations. The power of Smart-seq2 technology to cluster cell populations in comparison to microfluidic based technologies was highlighted in a recent study (Hagemann-Jensen M, et al. Nature Biotech 2020, Fig 1B below).

[REDACTED]

Moreover, in many previous data sets, there is a cross-contamination between different cell types that questions the value of that data for the scientific community (for an example see Fig 2 below). In our experience there is less risk for cross-contamination when cells are FACS-sorted before scRNA-seq (which is what we have done). To understand the true nature of glomerulus-associated cells, it is a major advantage to capture clean cell populations.

Fig. 2. Barplots showing the expression of the proximal tubulus (PT/PTC) marker *Slc34a1* in captured renal cell populations (First 2 rows: Sutzak et al. Science 2018; last row: our SMARTseq2 approach). In the first data set, the gene is detected in all kidney cell types

captured, demonstrating a massive cross-contamination of all captured cells by mRNA released from the abundant PT cells. In our data, the Slc34a1 signal is almost exclusively found in a PTC cluster and a subcluster of distal tubulus cells.

On the subject that smart-Seq affords deeper sequencing of each cell, this would make sense to state as an advantage if the study had indeed identified a novel pathway or receptor-ligand interaction not previously known, afforded to the authors by deeper sequencing. As it stands, this is not the case, so smart-seq2 sequencing has not resulted in new insights in this study.

The main novelty of our manuscript is that we have been able to identify several cell populations that have been missing in previous single cell studies, and pinpoint novel biology in these cell types. For instance:

-We have been able to identify true intraglomerular mesangial cells (MCs) (previously all PDGFRB positive cells were wrongly annotated as MCs). Our study unveiled the LRP1-mediated phagocytic machinery in these cells, and in the revised manuscript we provide evidence for this using *in vitro* and *in vivo* models (Fig. 3). Moreover, the precise identity of MCs unraveled a phenotype that is a combination of pericyte, vSMC and fibroblast.

-We identified extraglomerular MCs, a group of cells that has been unclear if it is really a unique cell type. We have been able to capture these cells and identify a molecular signature that will allow the isolation of these studies for future studies.

-We identified during the revision rare macula densa cells among the cTAL cell population and suggest the cytosolic 5'-nucleotidase encoded by NT5C1A as a new member of the macula densa adenosine signaling.

-We have for the first time captured PECs and demonstrate their global molecular signature in mouse.

Some specific comments:

Figures 1 and 2: all the markers identified in both mouse and human have been previously described by several other studies, so there is no new insights or conclusions drawn by these analyses.

We partially agree with this comment. We feel that showing an overview of how we have identified cell clusters is of importance for the reader to understand the results that are described later. This has been until now the standard way to present scRNA-seq data sets. In Fig1, we have added new results on active transcription factor regulons in single cells of three principal glomerular cell types (Fig. 1D-E). Fig. 2 has been re-organized as follows: i) data on GECs has been moved to supplements (Fig. S3); ii) data on PECs have been moved to Fig. 7, which now includes the activity of transcription factor Wt1 regulon in single cells from wild type and Pdgftb+ mice, as well as stainings for two new PEC marker proteins (LBP and DKK3). New Fig. 2 is now on the heterogeneity of MLCs.

Figures 4-6: the comparison of mouse and human podocytes, MCs and GECs is interesting, but quite similar to observations made by Harder et al. and others as above.

As the reviewer has described above, multiple scRNA-seq studies in either mouse or human kidney have been published. However, studies on systematic comparison of glomerular single cell profiles between mouse and human are missing. Based on literature search using "Harder et al and others", we have failed to find kidney scRNA-seq related publications performing mouse-human comparative analysis of three primary glomerular cells. To our knowledge, this comparison is to date the first one and provides many interesting insights into the translational difficulties. The major strength of this comparison is that we have used the identical cell isolation and sequencing platforms that makes the comparison more reliable. This is nicely highlighted from our in-silico validation of species-specific podocyte

genes using published bulk, single-cell RNA-seq data and also our in house single-cell data (Fig. 5d-5e)

The identification of Lacis cells as extraglomerular mesangial cells (also known as Polkissen cells, or Goormaghtigh cells) is interesting as a confirmation of past work showing the same.

Our study for the first time has captured these unexplored cells and provide their molecular signature. This is of crucial value as it:

- 1) Validates them as a unique cell type as they have a distinct molecular profile in comparison to intraglomerular MCs and VSMCs (Fig. 2)
- 2) Provides insights into their unknown biological role and makes it possible to trace them during disease progression. A very interesting question is for instance how they contribute to the increase of mesangial cell number within glomerular tufts in glomerulopathies. This can be analysed by performing SMARTseq2-based scRNA-seq in disease states.
- 3) Opens up a possibility to isolate them for cell culture studies. Previous studies with primary mesangial cell cultures have shown heterogeneity within these populations. Here below (Fig 3) we show an example from the study by Blom ID et al. (NDT 2001, PMID: 11390712) These cells were all defined as true MCs but it may be that they represent both intra- and extraglomerular MCs. Our molecular signatures allow the precise isolation of right cell types for downstream studies.

[REDACTED]

Reviewer #2 (Remarks to the Author):

He et al describe an elegant scRNA-seq study and reveal insights into the composition and heterogeneity of mesangial cells and some potential interspecies differences in glomerular gene expression patterns. Some concerns reduce enthusiasm for the manuscript. For example, the isolation methods likely introduce some differences in cellular composition between human and mouse samples. In addition, the numbers of cells derived (~800 from 8 human donors, ~1600 from 16 mice) seems low and raises concern for inefficient isolation and subsequent biased cell sampling/representation. Conversely, the number of genes recovered per cell is high compared to other scRNA-seq studies, increasing confidence in the differential expression patterns described by the authors. I think the strongest points of the manuscript are a) heterogeneity of mouse mesangial like cells with identification of a genomic fingerprint of Lacis cells and b) interspecies differences in podocytes. Additional protein-level validation of the key findings would greatly strengthen the authors' conclusions. This is especially necessary in the absence of specific mechanistic insights.

We agree that number of cells is low in comparison to other kidney scRNA-seq studies and the explanation is that we use Smart-seq2 method whereas others have used microfluidic-based methods. We argue that it is more important to have deep (i.e. a large fraction of the cells' transcriptomes covered) and clean (i.e. non-contaminated) data from relatively fewer

cells than to cover large numbers of cells that are contaminated or that do not include all important cell types or that small number of genes are detected. For instance, the first published scRNA-seq study of kidney tissue (Park et al, Science 2018) covered almost 58 000 renal cells. In spite of this, however, they failed to capture intra- and extra-glomerular mesangial cells, parietal epithelial cells of the glomerulus and macula densa cells. Outside of the glomerulus they also failed to capture renal fibroblasts (although they claimed to have done so). The lack of depth in that and other renal studies have hampered the identification of different renal subpopulations (see data below for comparison of depths).

We used Smart-seq2 platform, which is particularly suitable for the analysis of rare cell types, or any cells that are hard to capture in sufficient numbers for the conventional microfluidic-based methods. Smart-seq2 generates deeper data from each cell (Fig 4A below). Therefore, our choice of methodology is more feasible to identify and define rare cell populations. The power of Smart-seq2 technology to cluster cell populations in comparison to microfluidic based technologies was highlighted in a recent study (*Hagemann-Jensen M, et al. Nature Biotech 2020, Fig 4B below*).

[REDACTED]

Moreover, in many previous data sets, there is a cross-contamination between different cell types that questions the value of that data for the scientific community (Fig 5 below). In our experience there is less risk for cross-contamination when cells are FACS-sorted before scRNA-seq (which is what we have done). To understand the true nature of glomerulus-associated cells, it is a major advantage to capture clean cell populations.

Fig. 5. Barplots showing the expression of the proximal tubulus epithelial marker (PT) *Slc34a1* in captured renal cell populations (First 2 rows:: Sustzak-lab, Science 2018; lowest row: our SMARTseq2 approach). In the first data set, the gene is detected in all kidney cell types captured, demonstrating a massive cross-contamination of all captured cells by mRNA released from the abundant PT cells. In our data, the *Slc34a1* signal is almost exclusively found in a PT cluster and a distal tubulus subcluster.

As what comes to protein level validation, we have in the revised manuscript performed stainings/Westerns to validate some of our findings (Fig. 5G for podocytes, Fig. 7E for PECs and Fig. 8F for macula densa cells). Moreover, we have added functional data on endocytosis activity in intraglomerular mesangial cells (Fig. 3).

Specific comments:

- More information is required about how the 8 human donor kidney biopsies were collected. Were these patients undergoing a procedure (E.g. nephrectomy)? Was history of diabetes/hypertension an exclusion criterion? What was the age of the donors? Is the human data pooled from all 8 individuals (i.e. only ~100 cells/individual). In general, the numbers of cells available for analysis should be clearly stated at the outset in the results section.

Biopsies were from living related donor kidneys that were taken during the transplantation surgery (before the blood circulation was closed). These individuals had previously gone through an extensive clinical investigation to exclude diabetes / hypertension / other disorders that could potentially affect kidney function. Thus, they offer us an optimal “normal human kidney tissue”, especially when compared to traditional control materials used (nephrectomy kidneys). The information of the 8 living kidney donors (3 male and 5 female, mean age 48.2 ± 12 years old) in detail and the biopsy procedure has been described in Materials and Methods (p. 20, 2nd paragraph). Human data were pooled from 8 donor biopsies. Cell numbers for further analysis have been stated in Figure 1b.

- *Perhaps because so few cells were isolated, the authors do not appear to have identified populations of immune cell types such as dendritic cells or resident macrophages in their cell clusters despite a strong literature that such cells are present in the kidney.*

We actually identified a number of immune cell subtypes in our dataset. However, as immune cells were not the focus of this study, we did not present more data on them. As suggested by Reviewer 3 (see below), we have in the revision integrated the single-cell data of CD45⁺ cells from our study with the published immune cell atlas of mature human kidney from patients undergoing tumor nephrectomy by using Seurat v3 (Fig. S2). Our cell type annotations were in line with the immune cell atlas. We re-named our previous annotated “Monocyte” into “MNP” (mononuclear phagocyte), as we found that this cell population covers all identified MNP subpopulations in the reference immune cell atlas. The integrated results are presented in the revised Fig. S2.

- *The glomerular isolation methods are likely driving some of the cellular compositional differences between the human and mouse datasets. Serial sieving in human glomeruli usually strips off Bowman's capsule and therefore the lack of PECs is not surprising. Conversely, Dynabead assisted glomerular isolation in mice often retains Bowman's capsule thereby including PECs in the sample. This should be incorporated either into the methods or results section. More generally -- how much of the interspecies heterogeneity can be attributed to this difference in cellular sampling inherent to isolation?*

This is a very interesting comment. As pointed out by the Reviewer, the different methodology to isolate glomeruli in mouse/human is likely the reason why we do not capture PECs in humans. We have described in detail the isolation techniques used in Methods-section (p. 20, 3rd paragraph) and added a sentence to results section that clarifies this (p. 13, 1st paragraph).

Whether different glomerular isolation methods partially contribute to the interspecies diversity is a complex question. However, we captured the three principal glomerular cell types (podocytes, GECs and MCs) from both species, which were the main focus of this study. To minimize the possibility of species-specific transcriptomic differences driven by the isolation methods, we used the identical single cell dissociation protocol for both mouse and human glomerular fractions. To get further insights into this issue, we isolated in the revised manuscript mouse glomeruli using a new “non-bead based” method (described by Wang et al. Am J Physiol 2019) and performed scRNA-seq. After scRNA-seq of mouse glomerular cells prepared by “the non-bead based method”, the expression of species-specific genes was similar in podocytes/GECs as shown in Fig 5E and Fig 6C. This supports the idea that the isolation method does not affect the differences detected between mouse and human. Moreover, bulk RNAseq data shown in Fig 5D is nicely in line with our data and validates species differences.

- *Fig. 2a is superfluous. Fig. 2b should include Wt1 to compare with podocytes as the other major cell type expressing this marker. Indeed, the assertion that WT1 is a highly specific marker of parietal epithelial cells, expressed in line 124, is erroneous as it is often used as a preferential marker of podocytes. It is thought the expression of this marker in parietal cells may represent a transitional stage by which parietal cells could differentiate into podocytes in certain disease and regenerative states. In Fig. 2c, rather than just showing staining for known markers of PECs, what about validating some of the novel markers identified by scRNA-seq? For the IHC panel, a zoom in on a single glomerulus with capsule would be easier to interpret the staining pattern at the cellular level.*

As the reviewer points out, WT1 is expressed also highly by podocytes. We have re-phrased the text concerning WT1 (p. 13, 1st paragraph). Moreover, we have performed transcription factor regulon activity analysis (according to Reviewer 3's suggestion) that shows that WT1 is active in both podocytes and PECs. In addition, as suggested, we validated immunohistochemically two new PEC markers, LBP and DKK3, as well as modified figures (zooms) according to the suggestion. This new data is shown in revised Fig. 7.

- *Fig. 3 could benefit from localization of MLC1-4 using a panel of markers identified from the sequencing studies. The authors localize MLC-C1 (Lacis) and MLC-C4 (intraglomerular mesangial cells). Where do MLC-C2 and C3 cells localize in/around the glomerulus?*

We have validated the localization of MLC1-4 by stainings as follows: 1) MLC-C4 as intraglomerular mesangial cells using PDGFRA reporter line (Fig. 2C); 2) MLC-C1 as extraglomerular mesangial cells located at the stalk/base of the glomerular tuft (combination of ASMA/GATA3/PDGFRB stainings, Fig. 2D-E); 3) MLC-C2 as vSMCs of afferent/efferent arterioles with calponin-1 staining (gene name *Cnn1* in Fig 2B); 4) MLC3 as renin producing cells in the juxtaglomerular apparatus with staining for renin (Suppl. Fig. S7, this last part was not in the original version of the manuscript)

- *I would caution against expansive statements such as "we here identified for the first time pure transcriptional profiles of MCs". From Fig. 3f, the true mesangial cell (MLC-C4) dataset appears to be derived from *only* 17 cells - and that too isolated from multiple mice. Also, the range of the Fig. 3f's heatmap could be reduced (e.g. from -2 to 4) to show more of the data since there appear to be very cells with expression values above 4.*

We agree with this comment. To validate the mesangial heterogeneity and identity, it is necessary to make a replication using an independent dataset. In the revision, we sorted EGFP-labeled PDGFRB⁺ cells of isolated glomeruli from *Pdgfrb*-EGFP reporter mice followed by scRNA-seq. In this new dataset, the previous finding is fully replicated (additional 17 intraglomerular MCs) as shown in Fig S10.

As what comes to the comment for Fig 3F, the values are z-score and there are many cells with z-score above 4. Thus, we prefer to keep the original z-score values.

- *In Fig. 4a/5a/6a, highlighting some of the outlier and shared genes by callouts would be nice to see*

As suggested, we highlight top 15 most highly expressed species-conserved genes in the revised Fig 4A, 5A, 6A.

- *In Figures 4 and 6, validating Nfasc and Rxfp1 expression at the protein level would provide additional confirmation*

As suggested, we have validated the expressional difference between species for NFASC at the protein level in the revision (Fig. 5G). As what comes to *Rxfp1*, we failed to validate the finding at the protein level due to lack of good antibodies. We, together with our collaborators (AstraZeneca), have tested 5 different antibodies (Atlas, Santa Cruz, Abnova, Phoenix, Immunodiagnostik) for RXFP1. We used heart tissue as a known positive control but none of the antibodies gave reliable signal (in IHC and Westerns). Of note, for groups working with RXFP1 this is known to be very problematic. Instead, we have in situ hybridization data for *Rxfp1* in kidney tissue if the reviewer would wish us to include that in the manuscript.

- *I think the authors overstate the novelty of their finding that PLA2R is expressed by human podocytes and not those of mouse. This has been well recognized for some time by those who have been investigating the role of this protein in the pathogenesis of membranous nephropathy.*

We have revised the text according to the suggestion. Although we work with glomerular biology, membranous nephropathy has not been our focus and therefore we had missed this. We have revised the part of the discussion in the revised manuscript (taken PLA2R1 part out, p. 17, last paragraph).

- *The colors of the heatmaps vary across the figures. I understand that different measures are being represented, but some common ones (e.g. log2 RPKM) are shown in different color coordinates which is a bit jarring.*

As suggested, we have modified this by using a common color for the same units throughout in the revised figures.

- *A data sharing statement is not included in the manuscript. A portal to explore the data is now becoming standard for these types of studies and facilitates access to non-bioinformatically enabled researchers.*

We have added a data sharing statement in the revised manuscript (p. 30, last paragraph). All the raw data and processed data will be deposited to GEO, a common repository for sequencing data before publication. Moreover, we have built a user-friendly platform that makes our data easily usable for the scientific community: <https://patrakkalab.se/kidney> (now behind a password but open after publishing)

Reviewer #3 (Remarks to the Author):

General comments

The authors present a dataset of murine and human single-cell RNA sequencing data from kidney glomerular cell preparations using the Smart-Seq2 technology. This has allowed them to generate a high quality dataset with sensitive gene detection, which complements recently published broader and larger scale droplet based single-cell and single-nucleus RNA sequencing approaches of the mouse and human kidney 1–4. The authors report a fairly comprehensive inventory of glomerular cell types and their transcriptional profiles, and are able to recover expected cell types albeit with the conspicuous absence of glomerular parietal epithelial cells in the human data. In the paper's current form, the work describes a spatially resolved inventory of glomerular cells, but does not reveal and substantially novel biology which would merit publication in Nature Communications in its current form.

The paper currently lacks one or more features, which would significantly increase the impact of the paper such that it would be suitable for publication in Nature Communications:

Full "round trip" validation of identified cell populations - The paper is presented as a resource defining the "cellular identity of MCs". Whilst the authors show the expression of specific markers nominated from their scRNAseq data with microscopy, they then stop there. One approach would be to isolate cells on the basis of these markers using FACS, perform further scRNAseq, and illustrate that they recover the same transcriptionally distinct subsets with high fidelity. This approach has previously been employed in scRNAseq studies taking a similar focused approach.

First, we focused on PDGFRB⁺ cells as they had unravelled previously unknown heterogeneity. As suggested, we followed-up our first scRNA-seq data and used FACS to sort EGFP-labeled PDGFRB⁺ cells of isolated glomeruli for scRNAseq from *Pdgfrb*-EGFP reporter

mice. This generated an independent data set that fully replicated our previous findings and validates the mesangial heterogeneity (Fig. S10). In addition, we also captured a number of PDGFRB⁺ PECs, which confirmed the transcriptome profiles of previous annotated PECs and also the activity of Wt1 regulon in PECs (Fig. 1D). Moreover, we used FACS to sort podocytes for scRNA-seq using a reporter line expressing Tomato under a podocin promoter. Podocytes isolated by this way also replicated our previous findings on mouse-specific podocyte signatures (Fig. 5E).

Characterisation of functional properties of identified populations - as described extensively in the introduction, the glomerulus is “functionally and structurally unique” arrangement of cells. The specialised function of this apparatus relies on interactions and signalling between cell types. Mesangial cells for example, have been ascribed numerous functions, for example: phagocytosis; roles as pericytes, extracellular matrix production; and for “Lacis cells” signalling in tubuloglomerular feedback. It would be interesting to build a richer picture of the interactions and functions of these cells, initially building from analysis of the scRNAseq data, but ultimately with some validation and functional assays. Which should build on hypotheses generated from investigation of the data. For example, do the mesangial cell populations express phagocytic machinery? If so, experiments to validate this function would be valuable.

This is a very good comment. First, to highlight interactions and signaling between different glomerular cell types, we have performed ligand-receptor cross-talk analysis and the results are shown in revised Fig. 3A-B. Importantly, this is the first time that real intraglomerular MCs have been identified (unlike previous studies that categorized all PDGFRB⁺ cells as MCs), and thus shows most likely interactions that are more relevant. Second, to get insights into the function of intraglomerular MCs, we have analyzed their RNA profile in detail and can show that they have a clear fibroblast-like molecular fingerprint that links them to extracellular matrix production (p. 8, 2nd paragraph; and discussion in p. 15, 2nd paragraph). Interestingly, the transcriptome of true MCs unravelled also the presence of phagocytic molecular machinery, such as the presence of LRP1 that can potentially interact with endothelial F8/APP. LRP1 is a known multifunctional scavenger binding to multiple ligands and involved in receptor-mediated phagocytosis [Lillis AP, et al. *Physiol Rev* 2008;88:887]. To validate this functionally, we performed phagocytosis assays in isolated mouse glomeruli and could show that PDGFRB⁺ cells have this activity (Fig. 3d). Mechanistically, this seems to be mediated by LRP1 as a LRP1-antagonist effectively inhibited phagocytic activity (Fig. 3d). We demonstrated that phagocytosis is occurring also in human Pdgfrb⁺ glomerular cells (Fig. 3e). Finally, intravenously injected FITC-labelled BSA was taken in by mouse MCs, suggesting that this may occur also in vivo (Fig. 3f). Together, the data provides a critical novel insight into the function of MCs in the glomerular filtration barrier.

Delineating conserved transcriptional programs at play between murine and human glomerular cell subsets in glomerular disease settings - Throughout the introduction and discussion the authors highlight the translational significance of understanding the molecular composition of glomerular cell types. Ideally the authors would extend their study by adding such translational significance. Kidneys from murine glomerular disease (eg diabetic injury, nephrotoxic nephritis etc.) could be compared to biopsies from kidneys with cognate human disease to uncover conserved glomerular cell type specific transcriptional patterns in disease - for example activation state associated with mesangial expansion in diabetic glomerular injury.

We agree that this is of critical importance. We are currently profiling human renal disease biopsies and mouse models of glomerular diseases using both single cell and single nuclear

RNAseq. However, these are extensive efforts and we think that they should be reported in separate publications.

Specific comments

Number of cells: Lines 40-42: "To this end, we used Smart-seq2 to profile large numbers of individual glomerular cells isolated from human living donor renal biopsies and mouse kidney" - the authors should clarify the number of cells they analysed. How many cells failed quality control? Each legend should clearly state the number of cells analysed in each UMAP plot.

We did scRNA-seq on 3938 mouse and 2188 human single cells using unbiased FACS sorting. After stringent quality control, 2277 mouse and 766 human high quality single cells were remained in the analysis. For single cells from C57BL/6 mice with enriched sorting strategy and single cells from Pdgfrb+ mice, 600 out of 1152 and 344 out of 384 passed the quality control and were used for analysis, respectively. These numbers can be found in Fig 1B, Fig S6B and Fig S10B.

Reference for glomerular disease prevalence: Line 81-82: "Glomerular disease processes are a major clinical challenge as they account for 85% of cases leading to end-stage renal disease" requires a reference. Presumably the authors are happy to include secondary glomerular injury as a result diabetes mellitus and hypertension rather than only primary glomerulonephritides in their quoted prevalence estimate.

It is true that in this number we had included also systemic diseases that cause glomerular injury. We have re-phrased the sentence to clarify this and added a reference (p. 4, 2nd paragraph)

Previous report of murine mesangial cells: The authors should cite a study from 2017 which performed scRNAseq on murine mesangial cells using the Fluidigm C1 platform⁶.

We have added this reference to the manuscript (ref. 8).

Human glomerular parietal epithelial cells: As stated above, the absence of these cells in the human data is a clear weakness of the study, and should be addressed by generating additional human data to capture these cells before publication. This will allow the authors to extend their report on species conservation and differences between glomerular cells to glomerular parietal epithelial cells. Adding additional human data will substantially increase the value of this dataset as a resource.

As what comes to PECs, our study is the first one to report the transcriptomes of PECs in mouse tissue. We have unfortunately not been able to capture them for scRNAseq analysis in human kidney tissue despite major efforts. There may be several reasons for this:

-Glomerular isolation methods differ between mouse and man, where Bowman's capsules could be lost during sieving procedure. We have made efforts to modify the protocol for human glomeruli to capture PECs, for example, by use of only one 300- μ m sieve that allows glomeruli with capsules to pass through, but have failed. In addition, we have tried to micro-dissect glomeruli in a way that Bowman's capsules would be intact together with glomeruli, but still we have not been able to sequence PECs.

-It is possible that we capture PECs but they are more sensitive than other glomerulus-associated cells and therefore fail in QC of cells.

-It is possible that we capture PECs but only in minimal numbers. As we gain only a limited number of glomeruli in an individual biopsy (normally only 10-20 glomeruli per biopsy), it may be that only common cells are sorted.

Unclassified & unknown cells:

Fig 1B+C both show populations of unclassified cells. Fig S2A suggests that these are cells with low number of genes detected. Are these cells unclassifiable because they are of low quality? The authors should therefore consider removing these from the analysis or reviewing their quality control pipeline given that unclassifiable, low quality cells have made their way into the final dataset.

These plots also show clusters of cells which are annotated as “unknown”. In contrast to the “unclassified” clusters, these appear to have a large number of genes detected. The authors should attempt to annotate these cells, at least to a broad lineage.

We have re-analysed this data during the revision (clusters named as “unknown” and “unclassified”). “Unknown” cluster of cells shows expression of multiple cell type markers and double number of genes detected. Thus, we consider them to be likely doublets or highly contaminated cells (similarly to previous studies, see our response to reviewer 1 with a contaminated data from Park et al. Science 2018). Thus, we have removed these cells from the analysis.

Unclassified cells show clearly less genes detected per cell than other cells. The numbers are still clearly better than previous scRNA-seq data generated through drop-seq based approaches. However, in comparison to our Smart-seq2 data, they are of poorer quality and therefore we have removed them from the analysis as well.

Murine data structure: An initial murine scRNAseq dataset is presented in Fig 1A, however on lines 147-149, the authors write: “To address this, we first increased the number of Pdgfrb+ cells by sequencing cells from 6 additional mice (Fig S3a-b) and then pooled and reclassified all Pdgfrb+ MLCs..” This implies that the murine data is fragmented into two datasets. The authors would have improved power to detect distinct cell populations in their murine data if they combined these data in the initial analysis (Fig 1B).

This is a suggestion that we actually tried to do already earlier. We can see the value of pooling the data from all experiments. However, we lost some clear clusters from multi-dataset alignment. This may result from differences in the gene detection rate from cells captured by biased CD45⁻ CD31⁻ sorting and by the unbiased sorting experiments. Therefore, we prefer to keep them separate and use the dataset from biased CD45⁻ CD31⁻ sorting as an independent experimental validation with a focus on enriched cell populations such as Pdgfrb⁺ MLCs. In this way, both datasets are not batch corrected which can retain their original information.

Quantitative assessment of similarity to other datasets: The authors make claims of similarity or dissimilarity to recently published datasets which are well supported by quantitative/statistical analyses - for example Fig 3G+H.

Immune cell marker genes:

Supplementary figure 1E+F uses the following genes to distinguish immune cell subsets: CD68, CD79A, CD79B, CD3D, CD3G, KLRK1, KLRD1, CSF3R. Can the authors clarify why they selected these genes in particular? Whilst CD3D and CD3G are good markers for T cells, CD68 is a tissue macrophage marker and not typically used as a monocyte marker, CSF3R can be expressed on granulocytes and monocytes and so is not a sensible choice for distinguishing these specifically.

Suggested sets of genes commonly used for distinguishing these cell subsets are: CD4 T cells, CD3D, CD3G, TRAC, CD4; CD8 T cells, CD3D, CD3G, TRAC, CD8A, NKG7; B cells, MS4A1, HLA-DRA, CD19, HLA-DRA; monocytes, CD14 (classical), FCGR3A (non-classical), FCN1, VCAN, LYZ; neutrophils, FCGR3B, S100P, VCAN, NK cells; PRF1, NKG7, FCGR3A, NCAM1, KIR family genes.

Regarding annotation of immune cells, we chose the original CD markers from BD Bioscience (www.bdbiosciences.com/documents/cd_marker_handbook.pdf), as they were well established cell surface markers at the protein level used in FACS to isolate different subpopulations of immune cells. We agree with the reviewer that these were not the optimal choice for single cell transcriptome data, especially when the resolution of molecular signatures is increasing with scRNA-seq.

Here below (Fig 6) we show the expression of suggested immune cell markers in integrated data from cells of the kidney immune cell atlas and our mouse and human immune cells. Some of the markers were not specifically expressed in one cell type. For example, CD4 expressed in both CD4 T cells and MNPs; HLA-DRA expressed in both B cells and MNPs. Thus, for cell types without clear cell type specific markers, a combination of markers may help distinguishing each population. Based on the data, we have chosen a panel of markers that we show in revised Fig S2 to annotate the subpopulations of immune cells.

Fig. 6. The expression of suggested immune cell markers in data from cells of the kidney immune cell atlas and our mouse and human immune cells.

Of note, for neutrophils we propose to use CSF3R as a marker as it is highly expressed by neutrophils (see Fig 7 below and Fig 6 above)

Fig. 7. CSF3R RNA expression in human blood cells (data from *The Human Protein Atlas*. www.proteinatlas.org)

The authors may wish to utilise a recently published scRNAseq analysis of immune cells in the kidney to aid their classification of these cells 2, making qualitative or quantitative reference to this dataset. The authors should display the single cell data rather than just boxplots.

As suggested, we integrated the single-cell data of CD45⁺ immune cells from our study with the published immune cell atlas of mature human kidney nephrectomy [Stewart BJ, et al. *Science* 2019;365:1461] by using Seurat v3 (revised Fig. S2). Our cell type annotations were in line with the immune cell atlas. We corrected our previous annotated cell type name “Monocyte” into “MNP” (mononuclear phagocyte) as we found this cell population covers all identified MNP subpopulations in the reference immune cell atlas, including mostly classical monocytes, non-classical monocytes, tissue macrophages and dendritic cells. Since immune cells are not the focus of this study, we prefer to annotate the main cell types instead of cell type sub-populations.

The result further confirmed our cell type annotation (revised Fig. S2B) in a quantitative way. Moreover, based on the cell type markers suggested by the reviewer and the integration of both human and mouse data, we visualized their expression at a single cell resolution of the integrated dataset and identified a good cell type marker panel as described above (revised Fig. S2C). The expression of these markers was also visualized with violin plots for each species separately based on original RPKM values (revised Fig. S1E-S1F).

Juxtaglomerular apparatus: It is surprising that the authors were unable to recover macula densa tubular cells given that they isolated glomeruli. Can the authors comment on why they think these cells were not recovered? Are they in fact within the populations shown in Fig1A+B but not correctly annotated?

This is a very interesting question as nobody has so far been able to capture rare macula densa epithelial cells (MDCs) for scRNA-seq analysis. Based on this suggestion, we re-analyzed the cTAL/DCT2 cell populations, since MDCs are physically located in the region of cTAL/DCT with high transcriptomic similarities. Re-clustering actually revealed a small cell population (n=11) in the vicinity of cTAL and we annotated them as MDCs because several known classical macula densa marker genes (*Slc9a2*, *Ptgs2*, *Slc12a1* and *Nos1*) were up-regulated in these cells, compared with other tubular cell groups. In the revision, we have re-annotated all tubular cells in UMAP as shown in Fig. 8a. Interestingly, this resulted in the identification of a novel molecular pair in MDCs, a cytosolic nucleotidase NT5C1A catalyzing the conversion of AMP to adenosine and a corresponding adenosine transporter SLC29A1. We further confirm NT5C1A as a new MDC marker immunohistochemically (Fig 8F). Due to the novelty, we present MDCs as a separate figure panel (Fig 8).

Collagen type 4: Type 4 collagen form an essential component of the glomerular basement membrane and is affected in Alport disease and anti-GBM (Goodpasture's) disease. This study permits a unique opportunity to understand which cells produce this important feature of the filtration apparatus. Can the authors comment on the expression patterns of collagen type 4 genes? Can the authors clarify further why they instead chose to focus on COL6A1?

In our scRNAseq generated by SMARTseq2 approach, COL4A1/2 genes are expressed by glomerular endothelial and mesangial cells, whereas Alport Syndrome associated COL4A3/4/5 genes are expressed only by podocytes (Fig. 8 below). This highlights the value (and purity) of our data set. The reason we in this case showed a poorly characterized glomerulus-expressing collagen COL6A1 is that this gene showed species-specific expression pattern: high expression in human mesangial cells but absence/very low expression in mouse mesangial cells. COL4 genes showed similar expression distribution in both mouse and man.

Fig. 8. COL4A1/2 genes are expressed by glomerular endothelial and mesangial cells, whereas Alport Syndrome associated COL4A3/4/5 genes are expressed only by podocytes in our scRNAseq.

Description of mesangial cell phenotype: Lines 194-195: "...suggesting that MCs possess as complex phenotype that is a combination of pericytes, vSMCs and fibroblasts". The authors should tone down this statement and rephrase along these lines: "suggesting that MCs possess a complex phenotype with features of pericyte, vSMC, and fibroblast transcriptional programmes".

We have re-phrased this statement according to the reviewer's suggestion.

Transcription factor networks: There would be substantial value added to the paper through attempting to understand the transcription factor networks defining each cell type. This might be straightforwardly accomplished using the Pyscenic set of computational tools7 - <https://github.com/aertslab/pySCENIC#id5>. This would help to address important questions this study leaves unanswered such as:

Given PEC and podocytes express WT1, are WT1 transcriptional targets active in both cell types?

This was a very valuable comment and we have done an extensive analysis of transcription factors during the revision. We applied the SCENIC method to study gene regulation networks in our mouse and human glomerular single cell data. The UMAP on binary regulon activity colored by our pre-defined cell type annotation has been added (Fig. 1D and Fig. S5). We focused on principal glomerular types (GEC, podocytes and MCs) and identified a set of

cell type specific active regulons common between human and mouse (Fig. 1D), for example *Wt1* in podocytes, *Gata3* in MLCs and *Erg*, *Gata2*, *Gata5*, *Sox18*, *Rarg*, *Pbx1* and *Bcl6b* in GECs.

As suggested, we also analyzed WT1 in detail. *Wt1* regulatory module appeared to be active in both human and mouse podocytes. As we previously saw *Wt1* also expressed in mouse PECs, we analyzed its regulon activity in these cells. Interestingly, *Wt1* transcriptional targets were also active in PECs, although the activity was clearly stronger in podocytes (Fig. 7B).

The authors argue that mesangial cells exhibit a phenotype that is a “combination of pericytes, vSMCs and fibroblasts” - is this determined by combinations of transcription factors targeting the same genes as in these cells?

To address this question, we performed regulon activity analysis on pericytes, vSMCs, FBs from the lung single-cell data [Vanlandewijck M, et al. Nature 2018;554:475] using SCENIC. This data set has been generated using SMARTseq2 protocol, identically to our study. Next, we integrated the regulon activity data from both human and mouse MCs and compared to lung pericytes/vSMCs/FBs. As shown in the revised manuscript Fig. 2I, mesangial cells shared several active regulons specific to pericytes, vSMCs and FBs. There are some differences between human and mouse MCs (Fig 2I). This is in line with the pattern we saw from the prediction in Fig. 2H. Taken together, MCs could be determined by combinations of TF regulons specific to pericytes, vSMCs and FBs.

Intercellular communication networks: It would be revealing to understand the putative cell-cell interaction networks between glomerular cell types. This could be accomplished using tools such as CellphoneDB 8 or NicheNet9.

We have performed in-depth analysis of ligand-receptor interactions across GEC, podocytes and MCs using the NicheNet ligand-receptor database. The results are presented in the revised Fig. 3B. The expression of these cell type specific ligands and receptors are also shown in revised Fig. S12.

*Gene names: Gene names should be italicised, and the authors should check the manuscript to ensure that they are using the correct capitalisation for murine vs human gene names, one such example is line 223 - “Violin plots of *Nfasc*, as an example of a human-specific gene, demonstrated a robust expression in human podocytes and its absence in mouse”.*

We have gone through the manuscript and corrected gene/protein names.

Mesangial cell relationships - can the authors clarify further the relationship between the intra- and extra-glomerular mesangial cells they find. There is some effort made at implying a trajectory between these cells (Fig3A), so can this be extended by some determining some directionality (for example using RNA velocity10)? It would be very interesting if these cells interconvert.

We have considered the option of running Velocity analysis of MC subpopulations. However, the interpretation would be difficult because the data we have is from two different batches and we need corrected data. Therefore, we have decided not to include that in our manuscript.

Data availability: The data must be made available in raw and processed annotated form, ideally via a user friendly portal. This will maximise the utility of this data for the kidney research community. We would encourage the authors to submit their data to the Human Cell Atlas data coordination platform. There is no comment on data availability in the manuscript.

We have added a data sharing statement in the revised manuscript (p.30, last paragraph). All the rawdata and processed data will be deposited to GEO, a common repository for sequencing data. Moreover, we have built a user-friendly platform that makes our data easily usable for the scientific community: <https://patrakkalab.se/kidney> (now behind a password but open after publishing)

Code availability: The code used in the analysis should be published on github or similar as is standard in the field. There is no comment on code availability in the manuscript.

Supplementary data format: Supplementary data files should be provided as .csv not .xls files. There are well reported issues with corruption of gene names in .xls files 11.

Analysis jupyter notebooks will be available in github.

References

1. Park, J. et al. Comprehensive single cell RNAseq analysis of the kidney reveals novel cell types and unexpected cell plasticity. doi:10.1101/203125.
2. Stewart, B. J. et al. Spatiotemporal immune zonation of the human kidney. Science 365, 1461–1466 (2019).
3. Young, M. D. et al. Single-cell transcriptomes from human kidneys reveal the cellular identity of renal tumors. Science 361, 594–599 (2018).
4. Lake, B. B. et al. A single-nucleus RNA-sequencing pipeline to decipher the molecular anatomy and pathophysiology of human kidneys. Nat. Commun. 10, 2832 (2019).
5. Villani, A.-C. et al. Single-cell RNA-seq reveals new types of human blood dendritic cells, monocytes, and progenitors. Science 356, (2017).
6. Lu, Y., Ye, Y., Yang, Q. & Shi, S. Single-cell RNA-sequence analysis of mouse glomerular mesangial cells uncovers mesangial cell essential genes. Kidney Int. 92, 504–513 (2017).
7. Aibar, S. et al. SCENIC: single-cell regulatory network inference and clustering. Nat. Methods 14, 1083–1086 (2017).
8. Efremova, M., Vento-Tormo, M., Teichmann, S. A. & Vento-Tormo, R. CellPhoneDB v2.0: Inferring cell-cell communication from combined expression of multi-subunit receptor-ligand complexes. doi:10.1101/680926.
9. Browaeys, R., Saelens, W. & Saeys, Y. NicheNet: modeling intercellular communication by linking ligands to target genes. Nat. Methods (2019) doi:10.1038/s41592-019-0667-5.
10. La Manno, G. et al. RNA velocity of single cells. Nature 560, 494–498 (2018).
11. Ziemann, M., Eren, Y. & El-Osta, A. Gene name errors are widespread in the scientific literature. Genome Biol. 17, 177 (2016).

REVIEWER COMMENTS

Reviewer #2 (Remarks to the Author):

The authors have made a serious attempt to address the reviewers' concerns, including mine. The main issue, which all the reviewers brought up, were the small number of cells in the assay, which could introduce bias. The authors counter that the SmartSeq2 method is better suited for deep profiling vs. surveying a large number of cells. This is a reasonable contention, albeit open to dissent. Having a deep dataset is very useful for discovery. It does provide valuable counterbalance to the plethora of sc/snRNA-seq papers which are shallower in # genes detected per cell. The deeper characterization of mesangial cells is valuable – the level of validation appears appropriate for an initial description. They have toned down some of the expansive statements. They have also attempted to address the isolation differences between mouse and human – though this may still be playing a major role in the types of cells they are extracting.

Reviewer #3 (Remarks to the Author):

He et al. Single-cell RNA sequencing reveals the mesangial identity and species diversity of glomerular cell transcriptomes

GENERAL COMMENTS

He et al. have returned a revised manuscript in light of the reviewers' comments.

In our original remarks, we suggested that to increase the impact of the paper such that it would be suitable for publication in Nature Communications, the authors would need to perform one or more of the following:

1. Full "round trip" validation of identified cell populations
2. Characterisation of functional properties of the identified populations
3. Delineating conserved transcriptional programs at play between murine and human glomerular cell subsets in glomerular disease settings.

The authors have delivered new data on points 1 and 2.

1) Full "round trip" validation of identified cell populations - the authors have provided data from Pdgfrb+ sorted glomerular cells from a Pdgfrb-EGFP reporter mouse and present this data in figure S10 (344 cells) which recapitulates the populations they described in Fig 2. These data are a useful addition to the overall dataset, but do not validate markers generated from analysis of this dataset. Likewise sorting podocin+ cells, unsurprisingly yields podocytes. These experiments expand the data presented and support the existing conclusions, they do not perform a "round trip" validation of subpopulations (eg mesangial cells, extraglomerular MCs) per se - which would link transcriptional states to protein level data.

2) Characterization of functional properties of the identified populations - the authors have provided data claiming to support a role for mesangial cells in phagocytosis via the scavenger receptor LRP1. Here the authors have performed a ligand-receptor crosstalk analysis using the nichenet tool, and focus on the LRP1-F8/APP interaction showing co-localisation of LRP1 and PDGFRB in the mouse glomerulus. They then provide data claiming that mesangial cells utilise LRP1 for phagocytosis and that an LRP1 antagonist blocks this, using an ex vitro bead-based assay and flow cytometry read out. There are substantial issues with these data as outlined in the specific comments below.

OVERALL ASSESSMENT

- The scale of the data is small. I agree with R1&R2 that the limited scale of the data does mute enthusiasm for the study, however the authors raise reasonable responses regarding the quality of the data they present. The limited scale of the data becomes a real weakness where conclusions are drawn about rare populations (eg macula densa) from a handful of cells.
- Point 1 is minimally addressed; the authors provide a small amount of data sorting on

established markers for broad populations. Whilst these data validate the findings from the original dataset in as far as the population structures are recapitulated, it does not offer a means of enriching or sorting for these populations as thorough "round trip" experiments would (see Villani et al. 2018, Aizarani et al. 2019 for examples)

- The response to point 2 is potentially interesting, but there are serious shortcomings with the data presented - outlined in the specific comments.
- The absence of glomerular parietal epithelial cells in the human data remains a shortcoming and probably relates to data scale.
- In our opinion, the manuscript does not currently deliver a sufficiently complete set of data on either point (1 or 2) to merit confident recommendation for publication in Nature Communications.

SPECIFIC COMMENTS

1) Cell numbers - whilst the authors have clarified the cell numbers remaining in the final annotated dataset, it would be helpful if they could express how many cells are lost in QC in the appropriate methods section (line 555 onwards, "scRNA-seq data processing, quality control and filtering). I.e 600 out of 1152 and 344 out of 384.

2) Leukocyte annotation (figure S2) - the authors have tackled the annotation of their leukocyte subsets by jointly analysing these data with data from Stewart et al. 2019.

- Overall figure S2 is consistent with their annotation
- Their finding that CD4 is expressed on mononuclear phagocytes and CD4+ T cells is well known (at both transcript and protein level), likewise the HLA-DRA expression by both mononuclear phagocytes and B cells is canonical.
- The panel of markers used in Figure S2 is very sensible and nicely illustrates the populations identified.
- CSF3R is a previously described marker for neutrophils and it is very reasonable to use this.
- It is reasonable to annotate mononuclear phagocytes together.

3) Transcription factor networks - the authors have conducted a very interesting transcription factor regulon analysis and present this in Fig 1d and Fig S5.

- One minor comment: Fig 1d might be more readable if the transcription factors were ordered similarly on the mouse and human heatmaps, or if the cell type restricted regulons identified in both mouse and human could be highlighted.
- In the text, can the authors comment on any transcription factors identified in this analysis which were previously not described as associated with these cell types in the literature?

4) Unclassified cells - the authors have removed the "unknown" and "unclassified" cells determining them to be doublets or contaminated cells. This has resulted in a cleaner dataset.

5) Chord plots (Fig 3B) - whilst this is a well motivated way of presenting the data, it is unclear what the thickness of the chords corresponds to, and the distinction between ligands and receptors could be made clearer than the light/dark color scheme. As a further minor comment, can the titles for these plots ("human", "mouse") be moved above or below the plots rather than on top of them?

6) Ligand receptor analysis - can the authors clarify why they chose to focus on the LRP1-F8/APP interactions? Perhaps starting with the hypothesis that mesangial cells are phagocytic, then highlighting ligand-receptor pairs which might be implicated in this process would be a more principled way of determining which hit to follow up?

7) Murine glomerular cells phagocytosis assay - Figure 3D:

- This experiment appears to have been conducted with n=1 biological replicates, which is currently insufficient. n=3-5 with statistics across biological replicates is required for this sort of data.
- No mention is made of the statistical test employed in the table for Figure 3D
- The number of cells assayed by flow cytometry are insufficient to come to meaningful conclusions - they show very small numbers of bead+ cells.
- I cannot see a clear positive control for mCherry signal. This is made all the more important

because the fluorescence intensities presented are relatively dim.

For these data to convincingly support their claims, the assay would need: a) more biological replicates (3-5) b) an order of magnitude more cells per replicate.

8) In vivo phagocytosis (microscopy) - figure 3F. The authors perform an in vivo murine assay, injecting fluorescently labelled BSA, which co-localises with mesangial cells. Whilst interesting, the issues with these data are:

- Line 259-260, the authors claim that the FITC-labelled BSA accumulates in GATA3+ mesangial cells, but there is no GATA3 co-stain shown.
- Does not support the claim that this phagocytosis is dependent on LRP1 (for this in vivo administration of the LRP1 antagonist, or genetic ablation of LRP1 would be required).
- Again, appears to be n=1 biological replicates, which is currently insufficient.
- It is unclear why the authors opted for labelled BSA rather than latex beads as shown earlier in figure 3 for the experiment shown in figure 3F.

9) Figure 3 C & F - there are no scale bars. There is no colour legend, beyond a description in the legend - what is blue? (?hoechst).

10) Juxtaglomerular apparatus - the authors have identified 11 cells which express some macula densa marker genes (Slc12a1, Slc9a2, Ptgs2, Nos1) in their mouse data, and identify Nt5c1a as a marker of these cells, using human protein atlas data as validation. This figure is currently weak for the following reasons that should be addressed:

- The authors describe SLC29A1 in their response to the review, but this is not elaborated on in the figure (Fig 8) or in the text beyond including it in their dotplot.
- Given this finding is generated from their mouse data, the authors should provide immunohistochemistry of mouse kidney illustrating coexpression of NT5C1A and other markers of juxtaglomerular apparatus, ideally SLC29A1 given that they have highlighted the adenosine transporter and nucleotidase pair.
- The observation of juxtaglomerular cells, whilst potentially interesting, is very limited by the cell number (just 11 cells), so further validation is certainly required to confidently assert that these cells are indeed juxtaglomerular cells. A single marker in the human protein atlas data is currently insufficient. Ideally immunofluorescence staining of two or more markers identified in their single-cell data.

11) Human PEC - it is interesting that the authors were not able to isolate human PEC. It is likely that this is a result of rarity of the cells in the limited sample they have (renal biopsy), and they should be explicit that the lack of PECs in the human data likely results from rarity of the cell type in the face of limited starting material. An alternative explanation is that because these cells have a unique flattened morphology, they are not included in the gates the authors have used for sorting.

12) Collagen genes - the justification for exploring COL6A1 expression in the manuscript is reasonable.

13) Data sharing - data has been submitted to GEO and is available on a portal which we don't have access to at the moment as it is password protected - therefore we cannot comment on the content or quality of this. We would encourage the authors to upload their data to the human cell atlas, but this should not be an absolute requirement for publication.

14) Figure S13 - this is labelled as Fig S9

15) Figure S16A - the human data Y axis is labelled "GFP" - clearly this is not what is being measured.

16) Gene name style - Throughout gene names should be italicized (this is an issue chiefly in the figures)

Reviewer #4 (Remarks to the Author):

The authors have made a serious effort to substantially address the review comments on the original version of the manuscript, and the paper is now much improved.

REVIEWER COMMENTS

Reviewer #2 (Remarks to the Author):

The authors have made a serious attempt to address the reviewers' concerns, including mine. The main issue, which all the reviewers brought up, were the small number of cells in the assay, which could introduce bias. The authors counter that the SmartSeq2 method is better suited for deep profiling vs. surveying a large number of cells. This is a reasonable contention, albeit open to dissent. Having a deep dataset is very useful for discovery. It does provide valuable counterbalance to the plethora of sc/snRNA-seq papers which are shallower in # genes detected per cell. The deeper characterization of mesangial cells is valuable – the level of validation appears appropriate for an initial description. They have toned down some of the expansive statements. They have also attempted to address the isolation differences between mouse and human – though this may still be playing a major role in the types of cells they are extracting.

Thank you for the positive/constructive comment. Both shallower sequencing with high number of cells and deeper sequencing with fewer cells (our study) are essential as the data generated is very different using the two approaches so that neither of the methods can answer all biological questions that can be covered by combining the two platforms. Our data provides deeper profiling and therefore can identify cell types that have been missed in previous studies. As what comes to differences in isolation protocols between mouse and human, we agree that this can affect which cell types are captured for sequencing. However, our complementary studies suggest that the cells captured do not show alterations in transcriptomes that would be dependent on the isolation protocol, and thus comparison between mouse and human transcriptomes in captured cells is relevant in our study.

Reviewer #3 (Remarks to the Author):

He et al. Single-cell RNA sequencing reveals the mesangial identity and species diversity of glomerular cell transcriptomes

GENERAL COMMENTS

He et al. have returned a revised manuscript in light of the reviewers' comments.

In our original remarks, we suggested that to increase the impact of the paper such that it would be suitable for publication in Nature Communications, the authors would need to perform one or more of the following:

- 1. Full “round trip” validation of identified cell populations*
- 2. Characterisation of functional properties of the identified populations*
- 3. Delineating conserved transcriptional programs at play between murine and human glomerular cell subsets in glomerular disease settings.*

The authors have delivered new data on points 1 and 2.

1) Full “round trip” validation of identified cell populations - the authors have provided data from Pdgfrb+ sorted glomerular cells from a Pdgfrb-EGFP reporter mouse and present this data in figure S10 (344 cells) which recapitulates the populations they described in Fig 2. These data are a useful addition to the overall dataset, but do not validate markers generated from analysis of this dataset. Likewise sorting podocin+ cells, unsurprisingly yields podocytes. These experiments expand the data presented and support the existing conclusions, they do not perform a “round trip” validation of subpopulations (eg mesangial cells, extraglomerular MCs) per se - which would link transcriptional states to protein level data.

2) Characterization of functional properties of the identified populations - the authors have

provided data claiming to support a role for mesangial cells in phagocytosis via the scavenger receptor LRP1. Here the authors have performed a ligand-receptor crosstalk analysis using the nichenet tool, and focus on the LRP1-F8/APP interaction showing co-localisation of LRP1 and PDGFRB in the mouse glomerulus. They then provide data claiming that mesangial cells utilise LRP1 for phagocytosis and that an LRP1 antagonist blocks this, using an ex vitro bead-based assay and flow cytometry read out. There are substantial issues with these data as outlined in the specific comments below.

OVERALL ASSESSMENT

- The scale of the data is small. I agree with R1&R2 that the limited scale of the data does mute enthusiasm for the study, however the authors raise reasonable responses regarding the quality of the data they present. The limited scale of the data becomes a real weakness where conclusions are drawn about rare populations (eg macula densa) from a handful of cells.

As described in detail in our previous response, we argue that it is more important to have deep (i.e. a large fraction of the cells' transcriptomes covered) and clean (i.e. non-contaminated) data from relatively fewer cells than to cover large numbers of cells that are contaminated or that do not include all important cell types or that small number of genes are detected. As pointed out by the reviewer, the limited number of cells sequenced can be problematic when conclusions are made about small/rare populations, such as macula densa cells (MDCs) in our case. In the current revision, we have performed double immune-labelling experiments for MDCs and feel that we have convincingly shown that the identified population of cells represent true MDCs (Fig. 8f and 8g).

- Point 1 is minimally addressed; the authors provide a small amount of data sorting on established markers for broad populations. Whilst these data validate the findings from the original dataset in as far as the population structures are recapitulated, it does not offer a means of enriching or sorting for these populations as thorough "round trip" experiments would (see Villani et al. 2018, Aizarani et al. 2019 for examples)

As the reviewer points out, we used reporter mouse lines (Pdgrb+, Nphs2+) to isolate cells in the first revision, and could importantly recapitulate the subpopulations of cells identified in the original work. This was especially important when it comes to novel data on defining mesangial-like cells. We were not able to perform a thorough "round trip" of cell populations. The reason for this is that many of these cell types, especially mesangial-like cells, share most molecular features. For instance, in case of bona fide mesangial cells vs extraglomerular mesangial cells (Lacis), there is not a single gene that would be expressed at high levels in one of the cell types and be absent in the other one (Fig. S8). The molecular signatures are clearly different based on expression levels but no obvious markers that could allow us to sort these cells effectively for bulk studies was present.

- The response to point 2 is potentially interesting, but there are serious shortcomings with the data presented - outlined in the specific comments.

We have made a major effort to improve this data. For details see the response below and Fig. 3.

- The absence of glomerular parietal epithelial cells in the human data remains a shortcoming and probably relates to data scale.

- In our opinion, the manuscript does not currently deliver a sufficiently complete set of data on either point (1 or 2) to merit confident recommendation for publication in Nature Communications.

SPECIFIC COMMENTS

1) Cell numbers - whilst the authors have clarified the cell numbers remaining in the final annotated dataset, it would be helpful if they could express how many cells are lost in QC in the appropriate methods section (line 555 onwards, "scRNA-seq data processing, quality control and filtering). I.e 600 out of 1152 and 344 out of 384.

We have added this data to the revised manuscript (Lines 598-603).

2) Leukocyte annotation (figure S2) - the authors have tackled the annotation of their leukocyte subsets by jointly analysing these data with data from Stewart et al. 2019.

- Overall figure S2 is consistent with their annotation
- Their finding that CD4 is expressed on mononuclear phagocytes and CD4+ T cells is well known (at both transcript and protein level), likewise the HLA-DRA expression by both mononuclear phagocytes and B cells is canonical.
- The panel of markers used in Figure S2 is very sensible and nicely illustrates the populations identified.
- CSF3R is a previously described marker for neutrophils and it is very reasonable to use this.
- It is reasonable to annotate mononuclear phagocytes together.

Thank you for these comments. We agree with them.

3) Transcription factor networks - the authors have conducted a very interesting transcription factor regulon analysis and present this in Fig 1d and Fig S5.

- One minor comment: Fig 1d might be more readable if the transcription factors were ordered similarly on the mouse and human heatmaps, or if the cell type restricted regulons identified in both mouse and human could be highlighted.
- In the text, can the authors comment on any transcription factors identified in this analysis which were previously not described as associated with these cell types in the literature?

Thank you for the positive/constructive comment. We have modified Fig 1d according to the suggestion (see Fig 1d). Moreover, we have discussed the (previously unknown) presence of Nr2f2 and NR1H3 transcription factors in mesangial-like cells (Lines 149-152). Interestingly, both of these have been connected to renin regulation.

4) Unclassified cells - the authors have removed the "unknown" and "unclassified" cells determining them to be doublets or contaminated cells. This has resulted in a cleaner dataset.

5) Chord plots (Fig 3B) - whilst this is a well-motivated way of presenting the data, it is unclear what the thickness of the chords corresponds to, and the distinction between ligands and receptors could be made clearer than the light/dark color scheme. As a further minor comment, can the titles for these plots ("human", "mouse") be moved above or below the plots rather than on top of them?

The thickness and opacity of the arcs corresponds to the weight of ligand-receptor interactions from the NicheNet model. We have added a clarification for this to the figure legend (Fig 3b). Moreover, we have revised the chord diagrams in Fig 3b with more distinct colours for each cell type and ligand/receptor categories. As suggested, we have also moved the titles ("Mouse" and "Human") above each chord diagram.

6) Ligand receptor analysis - can the authors clarify why they chose to focus on the LRP1-F8/APP interactions? Perhaps starting with the hypothesis that mesangial cells are phagocytic, then highlighting ligand-receptor pairs which might be implicated in this process would be a more principled way of determining which hit to follow up?

As the reviewer points out, the reason why we mention this pathway is that phagocytosis by mesangial cells has been a controversial issue. To analyse this in a more unbiased way, we performed during the 2nd revision a pathway analysis for genes enriched in bona fide mesangial cells (vs other mesangial-like cells). This analysis pinpoints the enrichment of several pathways involved in phagocytosis in mesangial cells (Table S7). Based on this, we performed *in vitro* and *in vivo* assays to validate the phagocytic activity. We have clarified this approach in the revised results section (Lines 236-238).

7) Murine glomerular cells phagocytosis assay - Figure 3D:

- This experiment appears to have been conducted with $n=1$ biological replicates, which is currently insufficient. $n=3-5$ with statistics across biological replicates is required for this sort of data.

- No mention is made of the statistical test employed in the table for Figure 3D

- The number of cells assayed by flow cytometry are insufficient to come to meaningful conclusions - they show very small numbers of bead+ cells.

- I cannot see a clear positive control for mCherry signal. This is made all the more important because the fluorescence intensities presented are relatively dim.

For these data to convincingly support their claims, the assay would need: a) more biological replicates (3-5) b) an order of magnitude more cells per replicate.

Thank you for this very valid comment. To get enough biological replicates, we collected EGFP+ cells in 3 additional experiments (2-3 reporter mice per experiment, a total of 7 animals). Moreover, to increase the number of cells, we modified the record settings in collecting events of FACS and captured significantly more cells for analysis. The experiments clearly indicate the presence of phagocytic activity in mesangial cells (Fig. 3d). Additionally, we performed a similar bead-based phagocytosis experiment for confocal imaging, which nicely visualizes phagocytosed beads in partially digested glomerular EGFP+ cells and individual EGFP+ cells (Fig. 3e). As an independent positive control, we analysed the phagocytic activity of mouse blood leukocytes (Fig. S13b). These results, together with our other *in vitro* and *in vivo* assays, indicate that mesangial cells have phagocytic activity.

Of note, we discovered LRP1 (a well-studied scavenger receptor binding to a number of ligands for uptake via phagocytosis) in mesangial cells, and validated LRP1 expression in mouse mesangial cells immunohistochemically (Fig.S13a).

Experimentally, we made efforts to verify LRP1 as a receptor specifically mediating the phagocytosis in mesangial cells. The best way to do this would be to analyse glomerular cells from LRP1 KO mice.

However, the KO mice show embryonic lethality (Herz J, et al. Cell 1992) and are thus not feasible for analysis. Alternative option is to use an inhibitor. We used the only inhibitor available, receptor-associated protein (RAP, also called LRPAP1) in our assays. We failed to find reduced phagocytic activity of glomerular EGFP+ cells between RAP-treated and non-treated cells. In fact, RAP promoted phagocytosis in EGFP+ cells (see plot

below). Interestingly, RAP has previously been shown to promote uptake of a LRP1 ligand amyloid β (Kanekiyo et al JBC 2009;284:33352). It may be that in certain biological situations RAP promotes phagocytosis in a LRP1-dependent manner. However, it is also possible that RAP augments phagocytosis by promoting uptake of ligands in a LRP1-independent manner. As the interpretation of this LRP1-data is difficult, we do not want to include it in the final version of the manuscript. However, we feel that our complementary *in vivo* (see below) and *in vitro* assays demonstrate convincingly significant phagocytic activity in MCs.

Fig: *Effect of LRP1 antagonist RAP on uptake of latex beads by glomerular GFP-labeled PDGFRB+ cells. Proportion of bead+ cells treated with or without RAP in triplicate assays. Mean and SD is indicated (SD adjusted based on weight for cell numbers in each assay). The 2-sided P value was calculated using a proportion test.*

8) *In vivo phagocytosis (microscopy) - figure 3F. The authors perform an in vivo murine assay, injecting fluorescently labelled BSA, which co-localises with mesangial cells. Whilst interesting, the issues with these data are:*

- *Line 259-260, the authors claim that the FITC-labelled BSA accumulates in GATA3+ mesangial cells, but there is no GATA3 co-stain shown.*

We apologize for this mistake. We have corrected it to “Pdgrb+” mesangial cells (Line 264). To exclude Lacis cells, we only analysed cells in the periphery of the glomerular tuft for this assay (GATA3 labelling was not good enough for high resolution imaging).

- *Does not support the claim that this phagocytosis is dependent on LRP1 (for this in vivo administration of the LRP1 antagonist, or genetic ablation of LRP1 would be required).*

We agree with the comment. We have concluded that this data (together with *in vitro* and sequencing data) argues strongly for the presence of significant phagocytic activity in mesangial cells (Line 267-269). As mentioned above, LRP1 KO mice show embryonic lethality (Herz J, et al. Cell 1992) and are therefore not suitable for studying phagocytosis in mesangial cells.

- *Again, appears to be n=1 biological replicates, which is currently insufficient.*

We agree with this comment. We have repeated the experiment now a total of 3 times and the results are very clear: not a single Pdgrb+ cell in the glomerulus showed fluorescent signal in uninjected animals, whereas 70.0% (271/389) of analysed mesangial cells in three injected mice showed positivity for BSA (Fig. 3g)

- *It is unclear why the authors opted for labelled BSA rather than latex beads as shown earlier in figure 3 for the experiment shown in figure 3F.*

The reason for this is that we did not receive an ethical permit to inject latex beads to animals, whereas we received a permit to inject BSA.

9) *Figure 3 C & F - there are no scale bars. There is no colour legend, beyond a description in the legend - what is blue? (?hoechst).*

We have added the information to the revised manuscript (Fig. 3).

10) *Juxtaglomerular apparatus - the authors have identified 11 cells which express some macula densa marker genes (Slc12a1, Slc9a2, Ptgs2, Nos1) in their mouse data, and identify Nt5c1a as a marker of these cells, using human protein atlas data as validation. This*

figure is currently weak for the following reasons that should be addressed:

- The authors describe SLC29A1 in their response to the review, but this is not elaborated on in the figure (Fig 8) or in the text beyond including it in their dotplot.
- Given this finding is generated from their mouse data, the authors should provide immunohistochemistry of mouse kidney illustrating coexpression of NT5C1A and other markers of juxtaglomerular apparatus, ideally SLC29A1 given that they have highlighted the adenosine transporter and nucleotidase pair.
- The observation of juxtaglomerular cells, whilst potentially interesting, is very limited by the cell number (just 11 cells), so further validation is certainly required to confidently assert that these cells are indeed juxtaglomerular cells. A single marker in the human protein atlas data is currently insufficient. Ideally immunofluorescence staining of two or more markers identified in their single-cell data.

We agree that this figure needs more validation data due to the limited number of cells sequenced. We have performed the requested validation experiments and added a short comment on NT5C1A and SLC29A1 to the discussion (Lines 471-476).

We have added the following data to the revised manuscript:

- double labelling experiments with NT5C1A and NOS1 (known marker of macula densa cells) in mouse kidney tissue, and show that they co-localize to the same small subpopulation of tubular cells besides the glomerular tuft (Fig. 8f). Based on its localization and positivity for NOS1, this population represents macula densa cells.
- we have performed double labelling experiments with SLC29A1 and NOS1 in mouse kidney tissue, and show that also SLC29A1 localize to this group of tubular cells next to the glomerular tuft (Fig. 8f).
- we performed similar experiments in human kidney tissue and show that these two proteins co-localize to the same subpopulation of tubular cells in man (Fig. 8f).
- we have also added data from Human Protein Atlas showing that two different antibodies directed against NT5C1A stain similarly a small group of tubular cells next to the glomerulus (Fig. 8g).

We believe that this data significantly strengthens our conclusion that this population of cells represents true macula densa cells.

11) Human PEC - it is interesting that the authors were not able to isolate human PEC. It is likely that this is a result of rarity of the cells in the limited sample they have (renal biopsy), and they should be explicit that the lack of PECs in the human data likely results from rarity of the cell type in the face of limited starting material. An alternative explanation is that because these cells have a unique flattened morphology, they are not included in the gates the authors have used for sorting.

We agree with this comment. We have added a sentence to the result section for this (Line 341). We also believe that the reason is probably that the starting material is limited. As the reviewer points out, one other option is that we do not capture the cells due to gating settings. Another possible explanation could be that PECs are more sensitive to our dissociation protocol and get so damaged so that they do not pass QC.

12) Collagen genes - the justification for exploring COL6A1 expression in the manuscript is reasonable.

13) Data sharing - data has been submitted to GEO and is available on a portal which we don't have access to at the moment as it is password protected - therefore we cannot comment on the content or quality of this. We would encourage the authors to upload their data to the human cell atlas, but this should not be an absolute requirement for publication.

Raw sequencing and processed data of our mouse single cell cohorts and the

processed data of our human single cell cohort have been submitted to GEO repository. The accession number is GSE160048 and the data will be public available prior to the publication. Due to GDPR on human sensitive data, we have requested the submission of human raw sequencing data to EGA and our request is currently in process. In addition, we have made a website for the scRNA-seq data generated in this study. The address is <https://patrakkalab.se/kidney/> (user name: jakkop, password: glom). This website has a user-friendly interface that is feasible for non-bioinformatic users as well. Prior to the publication, this webpage will be made available for everyone.

14) Figure S13 - this is labelled as Fig S9

Thank you pointing this out, this has been corrected in the revised manuscript.

15) Figure S16A - the human data Y axis is labelled "GFP" - clearly this is not what is being measured.

To sort viable glomerular cells from human kidney biopsies, we stained dissociated single cells with a live cell dye cellTracker CMFDA-Green (Thermofisher) with excitation/emission spectra (492/517 nm) that is almost identical to alexa-488, FITC and GFP. This dye was also used for unbiased sorting of wild type mouse glomerular cells (Figure S17a upper panel). Thus, GFP-A was set on Y-axis during FACS sorting using BD AriaII or Melody. To avoid misunderstanding, we have changed Y-axis labelling to FITC-A.

16) Gene name style - Throughout gene names should be italicized (this is an issue chiefly in the figures)

This has been corrected in the revised manuscript.

Reviewer #4 (Remarks to the Author):

The authors have made a serious effort to substantially address the review comments on the original version of the manuscript, and the paper is now much improved.

REVIEWERS' COMMENTS

Reviewer #3 (Remarks to the Author):

The authors have made a significant effort to address our concerns. I think it would be reasonable to accept this manuscript.